# Yolk sac macrophage progenitors traffic to the embryo during defined stages of development

C. Stremmel [1,2], R. Schuchert[1,2], F. Wagner[1,2], R. Thaler[1,2], T. Weinberger[1,2], R. Pick[2], E. Mass [3,4], H.C. Ishikawa-Ankerhold[1,2], A. Margraf[2], S. Hutter[5], R. Vagnozzi[6], S. Klapproth[7], J. Frampton[8], S. Yona[9], C. Scheiermann[2], J.D. Molkentin[6,10], U. Jeschke [5], M. Moser[7], M. Sperandio[2], S. Massberg[1,2,11], F. Geissmann [3] & C. Schulz [1,2,11]

Tissue macrophages in many adult organs originate from yolk sac (YS) progenitors, which invade the developing embryo and persist by means of local self-renewal. However, the route and characteristics of YS macrophage trafficking during embryogenesis are incompletely understood. Here we show the early migration dynamics of YS-derived macrophage progenitors in vivo using fate mapping and intravital microscopy. From embryonic day 8.5 (E8.5) $CX_3CR1+$ pre-macrophages are present in the mouse YS where they rapidly proliferate and gain access to the bloodstream to migrate towards the embryo. Trafficking of pre-macrophages and their progenitors from the YS to tissues peaks around E10.5, dramatically decreases towards E12.5 and is no longer evident from E14.5 onwards. Thus, YS progenitors use the vascular system during a restricted time window of embryogenesis to invade the growing fetus. These findings close an important gap in our understanding of the development of the innate immune system.

[1] Medizinische Klinik und Poliklinik I, Klinikum der Universität, Ludwig-Maximilians-Universität, Marchioninistrasse 15, 81377 Munich, Germany. [2] Walter-Brendel-Center for Experimental Medicine, Ludwig-Maximilians-Universität, Marchioninistrasse 15, 81377 Munich, Germany. [3] Immunology Program, Sloan Kettering Institute, Memorial Sloan Kettering Cancer Center, 1275 York Avenue, New York, NY 10065, USA. [4] Developmental Biology of the Innate Immune System, LIMES-Institute, University of Bonn, Carl-Troll-Straße 31, 53115 Bonn, Germany. [5] Klinik und Poliklinik für Frauenheilkunde und Geburtshilfe, Klinikum der Universität, Ludwig-Maximilians-Universität, Maistrasse 11, 80337 Munich, Germany. [6] Department of Pediatrics, Cincinnati Children's Hospital Medical Center, 3333 Burnet Avenue, Cincinnati, OH 45229, USA. [7] Department of Molecular Medicine, Max Planck Institute of Biochemistry, Am Klopferspitz 18, 82152 Martinsried, Germany. [8] Institute of Cancer and Genomic Sciences, College of Medical and Dental Sciences, University of Birmingham, Edgbaston, Birmingham B15 2TT, UK. [9] Department of Immunology, The Weizmann Institute of Science, 234 Herzl Street, Rehovot 76100, Israel. [10] Howard Hughes Medical Institute, Cincinnati Children's Hospital Medical Center, 3333 Burnet Avenue, Cincinnati, OH 45229, USA. [11] DZHK (German Center for Cardiovascular Research), Partner Site Munich Heart Alliance, Biedersteiner Strasse 29, 80802 Munich, Germany. Correspondence and requests for materials should be addressed to C.S. (email: christian.schulz@med.uni-muenchen.de)

Macrophages are important effectors of innate immunity. They have a critical function in organ development, maintenance of tissue homeostasis and host protection during infection[1]. Furthermore, they have distinct functions in chronic inflammatory diseases such as atherosclerosis, metabolic disorders and cancer[2–5].

Historically, macrophages were believed to originate exclusively from bone marrow (BM)-derived monocytes after their recruitment to tissues. However, first macrophage progenitors develop in the YS, the first site of hematopoiesis in mammals[6]. Fate mapping analyses and lineage tracing experiments have indicated that tissue macrophages in many organs are of early embryonic origin[7–11]. Macrophages have been proposed to develop in the YS in two successive waves. First, primitive macrophage progenitors appear around E7.25 in the YS and predominantly infiltrate the brain after the onset of circulation[7,12–14]. Second, multipotent erythro-myeloid progenitors (EMP) develop from E8.25 onwards and are a major source of tissue resident macrophages[6,15,16]. Substantially later, starting from E10.5, hematopoietic stem cells (HSC) arise in the aorto-gonado-mesonephros (AGM) region and migrate to the liver to initiate fetal definitive hematopoiesis, which later shifts to the BM[17–19]. In adult life, monocytes and other short-lived myeloid cells are continuously replaced, whereas tissue macrophages can self-renew and persist independently of definitive BM hematopoiesis for prolonged periods of time, potentially throughout life[8,16,20]. However, even though macrophage trafficking during early embryonic development is a prerequisite for establishing this important component of the innate immune system, the kinetics of this process have not been defined.

EMPs are characterized by expression of the receptor tyrosine kinase KIT as well as the macrophage colony-stimulating factor 1 receptor (CSF1R). Subsequent differentiation into pre-macrophages is indicated by expression of the fractalkine receptor $CX_3CR1$, whereas mature tissue resident macrophages additionally express F4/80[12,21–23]. Thus, these markers allow genetic labeling and visualization of macrophage progenitors early during embryonic development.

In this study, we present in vivo data of EMP and pre-macrophage trafficking and proliferation in mouse YS and embryo. We identify a restricted time frame in which these cells travel from the YS to infiltrate the embryo via the bloodstream before becoming phenotypically mature macrophages within their tissue of residence.

## Results

### Intravital microscopy of YS macrophages.
To visualize the early development of macrophages in the mouse YS (Fig. 1a–f) and their trafficking to the embryo (Figs. 1g–k and 2a–f) we established an in vivo imaging strategy (Fig. 1b, c; Methods).

We first took advantage of $Cx_3cr1^{GFP/+}$ knock-in mice to visualize pre-macrophages and macrophages[21]. Green fluorescent protein (GFP)-labeling in $Cx_3cr1^{GFP/+}$ embryos corresponds to the CX3CR1+ CD45+ c-KIT- ("A3-like") population previously characterized in vitro, while other hematopoietic lineages are not labeled (Supplementary Fig. 1a–c)[8,12]. CX3CR1+ positive cells were not detectable in the YS by epifluorescence microscopy at E8.5; however, a rapidly expanding population of CX3CR1+ pre-macrophages appeared shortly thereafter (Fig. 1d). Macrophage numbers per area increased significantly in tissues during embryogenesis. Maximum cell density was approximately 100 cells per microscopic field (400× 400 µm) in the YS and reached a plateau by E12.5 with stable cell densities thereafter (Fig. 1d, e). In parallel to the increase in cell numbers, CX3CR1+ pre-macrophages underwent morphological changes from spherical

shape to mature cells with multiple dendrites (Fig. 1d) potentially engaging in direct cell-to-cell contacts, as it is typically found in adult tissues[24]. Moreover, pre-macrophages appeared partially associated with vascular structures (Fig. 1f and Supplementary Fig. 1d) and the first detection of CX3CR1+ pre-macrophages was temporally correlated with the formation of a dense vascular network (Fig. 1d, f and Supplementary Fig. 1d, e).

### Trafficking of YS macrophage progenitors via the bloodstream.
To date two alternate routes have been postulated of how macrophage progenitors enter the embryo, trans-tissue migration or trans-vascular trafficking via the bloodstream[7,13,22,25–27]. To address these two hypotheses, we directly imaged the trafficking routes of YS macrophage progenitors during embryogenesis using intravital microscopy. We observed CX3CR1+ pre-macrophages entering the YS vasculature (Fig. 1g, Supplementary Movies 1, 2) and traveling towards the embryo (Fig. 1h, i, Supplementary Fig. 1g and Supplementary Movies 3–5). The majority of pre-macrophages were trafficking in the YS bloodstream with an average velocity of about 210 µm/s, while some CX3CR1+ cells displayed slow surface translocation (rolling) at significantly lower mean velocity of 25 µm/s (Fig. 1j, k and Supplementary Fig. 1h). On occasion, pre-macrophages re-adhered transiently to YS endothelial structures after their vascular infiltration (Supplementary Movie 2). However, the biological meaning of these transient interactions remains unknown.

### Restricted time window of vasculature-mediated infiltration.
In vivo microscopy of the YS vasculature showed that CX3CR1+ cells appeared in the bloodstream shortly after the formation of a vascular network around E9 (Fig. 1i and Supplementary Fig. 1e)[7,28]. Subsequently, we observed a rapid increase in intravascular pre-macrophages trafficking to the embryo up to a maximum of about 30 cells/min at E10.5 in an average-sized vessel of about 60 µm in diameter at this developmental stage (Fig. 1i and Supplementary Fig. 1f, g). The highest number of pre-macrophages trafficking through the bloodstream per minute was observed between E9.5 and E10.5. At E12.5 the number dropped by more than 90% and no trafficking cells were observed at E14.5 (Fig. 1i, Supplementary Movies 3–5).

### Yolk sac macrophages in humans.
The process of hematopoietic development is conserved between mouse and humans[29]. In men, the YS serves as the initial site of hematopoiesis and gives rise to granulo-macrophage progenitors[30,31]. We collected fetal material from week 9 of gestation to determine the phenotype of human YS macrophages. YS macrophages were spherical as their mouse counterparts, while mature macrophages in the placenta presented the typical morphology of mature phagocytes with multiple dendrites (Supplementary Fig. 2a, b). Besides the human macrophage markers macrosialin (CD68) and epidermal growth factor module-containing mucin-like receptor 2 (EMR2, CD312)[32,33], macrophages of the human YS expressed CX3CR1 as in mice (Supplementary Fig. 2a, b). Thus, we identified similarities between mouse and human YS macrophages, opening the possibility of analogies in their development. The surface expression of EMR2 is of potential interest, since differential expression of EMR2 has been linked to macrophage maturation[33].

### Trafficking concurs with vascular network formation.
YS CX3CR1+ pre-macrophages are located in close proximity to vascular structures (Fig. 1f and Supplementary Fig. 1d), which is consistent with histological reports[34,35]. In the embryo, the primary target for YS-derived pre-macrophages was the developing brain, which is also characterized by the early establishment of a

vascular network (Fig. 2a–f and Supplementary Movie 6)[8,35]. Although some CX₃CR1+ cells were found in close proximity to the brain vasculature, they distributed throughout the head region forming a dense cellular network (Fig. 2b, e and Supplementary Movie 6). The infiltration of other tissues started thereafter reaching comparable cell densities of about 80 CX₃CR1+ cells/ microscopic field at E12.5 (Fig. 2b–d). Macrophage numbers in

YS and embryonic tissues remained stable from E12.5, supporting the notion that infiltration of the embryo by tissue macrophage progenitors was mostly completed at this time.

**Trafficking is restricted to spherical-shaped macrophages.** Next we addressed whether alterations in cell morphology

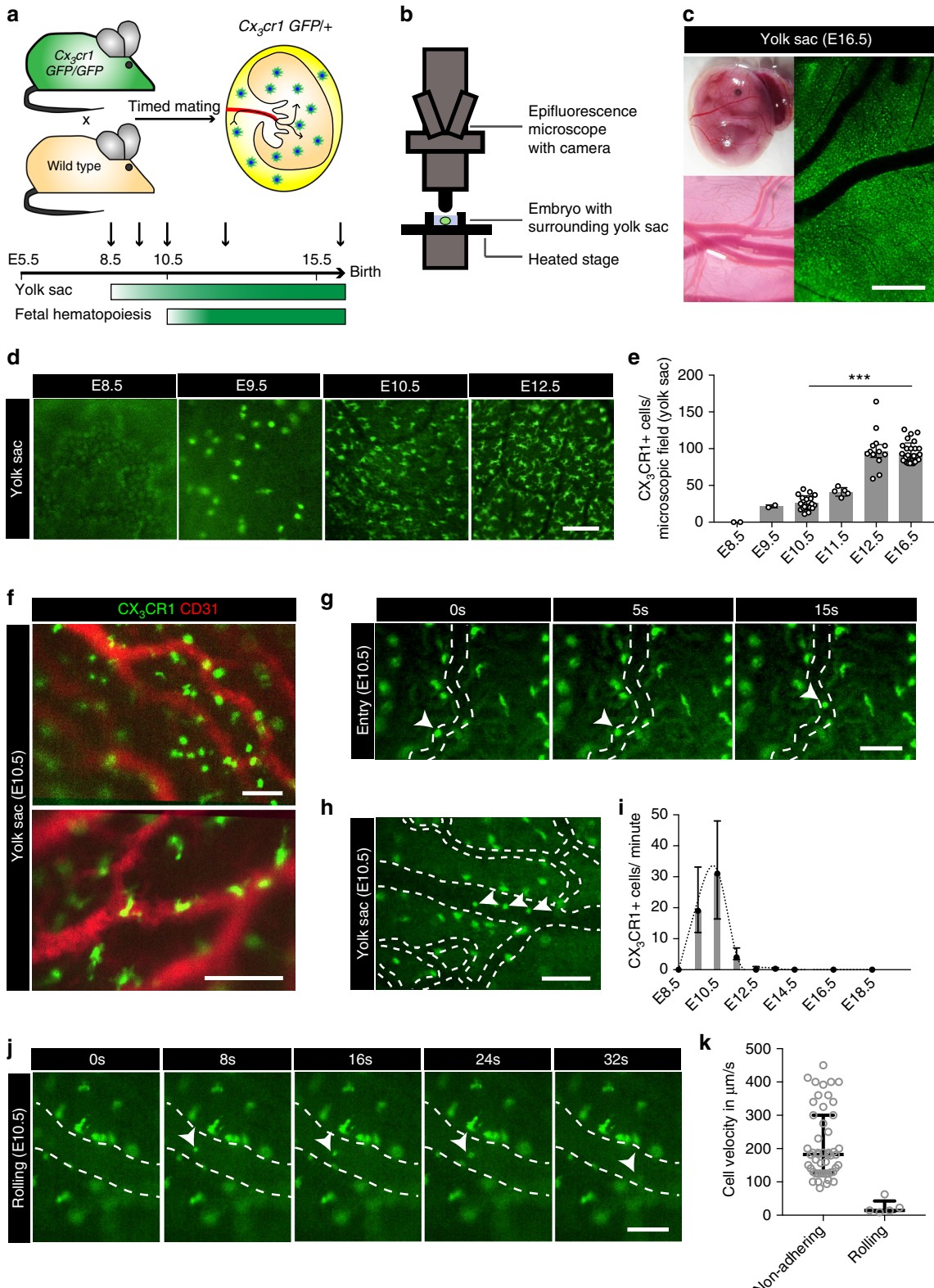

accompanying macrophage maturation were associated with macrophage trafficking. Pre-macrophages in the bloodstream displayed a spherical shape (Supplementary Movies 1–6). In contrast, dendrite-shaped YS macrophages did not enter the circulation or traveled through blood. The findings established with epifluorescence imaging were confirmed by high resolution spinning disc confocal microscopy in $Cx_3cr1^{Cre}$:$Rosa26^{mT/mG}$ embryos (Fig. 3a, b and Supplementary Movie 7). Thus, alterations in cell morphology correlate with restricted trafficking potential of YS macrophages. In line with this, large numbers of matured macrophages with dendrites remained in YS tissue after E12.5 (Fig. 1d, e, i). Interestingly, respective macrophages with mature morphology extended their dendrites into the YS tissue and also into the vessel lumen (Fig. 3c–f and Supplementary Movie 8).

**YS macrophage trafficking is independent of MYB and CX₃CR1.** Trafficking of pre-macrophages from the YS to the embryo took place mostly between E9.5 and E10.5, significantly decreased towards E12.5 and was not observed after E14.5 (Fig. 1i). This raised the possibility that pre-macrophage trafficking was restricted by the onset of fetal definitive hematopoiesis. HSCs arise in the AGM region from E10.5 and are found in the fetal liver by E12, where they produce hematopoietic cells[36]. We therefore revisited macrophage development in mice lacking fetal definitive hematopoiesis due to genetic absence of the transcription factor MYB (Fig. 4a, b). From E9.5 through E16.5 CX₃CR1+ cell numbers and densities in YS and embryonic tissues were not altered in the absence of HSCs and their progeny (Fig. 3c, d and Supplementary Fig. 3a–c)[8,22]. Using intravital microscopy, we found that numbers and timing of pre-macrophage trafficking from the YS to the embryo in $Myb$ knockout mice ($Myb^{-/-}$:$Cx_3cr1^{GFP/+}$) were comparable to wild type ($Myb^{+/+}$:$Cx_3cr1^{GFP/+}$) littermates (Fig. 4e and Supplementary Movies 9, 10). Thus, onset of fetal definitive hematopoiesis did not influence, or even restrict, pre-macrophage trafficking from the YS. These data also showed that most CX₃CR1+ cells identified in tissues after the onset of fetal hematopoiesis were derived from YS hematopoiesis but not from MYB-dependent hematopoietic progenitors. This is noteworthy, because CX₃CR1 is potentially expressed on various mature blood cells including fetal liver and BM monocytes[37], which could have biased our analysis at later stages of embryonic development (i.e. from E12.5). Further, our findings support the notion that YS-derived tissue macrophages are independent of the transcription factor MYB, which is consistent with previous reports[8].

The chemokine receptor CX₃CR1 mediates myeloid cell trafficking in inflammation[38–40] and macrophage colonization of the mouse embryo occurs in a CX₃CR1-dependent manner[21]. We therefore addressed whether pre-macrophage trafficking was modulated by CX₃CR1. In E10.5 embryos we found an increased cell density in the YS of $Cx_3cr1^{GFP/GFP}$ mice, while cell densities in embryonic tissues were decreased at this stage of development as previously reported (Fig. 4f)[21]. However, the number of cells

per minute trafficking from the YS to the embryo was similar between $Cx_3cr1^{GFP/GFP}$ (functional knockout) and $Cx_3cr1^{GFP/+}$ littermates (Fig. 4g). Further, the different hematopoietic lineages in the fetal liver were not altered by loss of one or both $Cx_3cr1$ alleles (Supplementary Fig. 3d–f). Thus, the molecular cues restricting macrophage migration from the YS to the embryo remain to be determined.

**Trafficking of CSF1R+ macrophage progenitors.** The major source of YS CX₃CR1+ pre-macrophages are EMPs[21], and we have previously shown that they can be labeled using a CSF1R-driven fate mapping system[16]. To define the trafficking dynamics of CSF1R+ macrophage progenitors we first performed intravital microscopy in $Csf1r^{Cre}$:$Rosa26^{eYFP}$ embryos (Fig. 5a). We observed a strong increase in YFP+ cells in YS and embryonic tissues during the early development with the spatiotemporal distribution pattern being comparable to that of CX₃CR1+ pre-macrophages (Fig. 5b–g). It should be noted that EMPs have multilineage (i.e. erythroid, megakaryocyte and myeloid) potential[23,41], resulting in the labeling of other hematopoietic cells in addition to macrophages[16]. Importantly, CSF1R+ cells traveled with similar velocity to CX₃CR1+ pre-macrophages in the YS vasculature and the maximum number of trafficking progenitors was reached between 9.5 and E10.5 in both models, thus showing similar trafficking dynamics (Fig. 5b–d and Supplementary Movies 11, 12).

While labeling of CSF1R-expressing cells is more efficient in mice with constitutively active Cre recombinase as compared to a tamoxifen (TAM)-inducible system, onset of fetal definitive hematopoiesis may lead to a bias during microscopy of YS macrophage trafficking. In order to label YS-derived CSF1R+ cells and their progeny before the onset of fetal definitive hematopoiesis[8], we crossed $Csf1r^{MerCreMer}$ mice with a $Rosa26^{eYFP}$ reporter and applied a single dose of 4-hydroxytamoxifen (OH-TAM) to induce expression of the yellow fluorescent protein (YFP) in a temporally-controlled manner (Fig. 6a–g). YFP labeling in E10.5 progenitors was low when OH-TAM was injected at E7.5, but increased about 6-fold when OH-TAM was injected at E8.5 (Fig. 6b, c), which is in line with previous reports that $Csf1r$ becomes active during maturation[12,16,34]. E8.5 OH-TAM injection labeled about 10–20% erythrocytes, granulocytes and macrophages whereas cells of the lymphoid lineage were not labeled, which is consistent with EMP multilineage potential (Supplementary Fig. 4a–d)[16,23].

YFP+ cells were first detected in the YS at E8.5 after OH-TAM pulse labeling at E7.5. Approximately 12 h later, coinciding with the onset of circulation, CSF1R+ cells could be also detected in the embryo (Fig. 6d). Trafficking kinetics of YFP+ cells in $Csf1r^{MerCreMer}$:$Rosa26^{eYFP}$ embryos OH-TAM-pulsed at E8.5 were comparable with that of $Csf1r^{Cre}$:$Rosa26^{eYFP}$ and $Cx_3cr1^{GFP/+}$ embryos analyzed earlier (Fig. 6g and Supplementary Movies 13, 14). In general, infiltration of the embryo occurred mostly between E9.5 and E12.5 in these mice confirming that the trafficking of macrophages and their progenitors from the YS via

**Fig. 1** Intravascular trafficking of CX₃CR1+ YS pre-macrophages. **a, b** Schematic graphs of the mouse model (**a**) and the intravital imaging setup in $Cx_3cr1^{GFP/+}$ mice (**b**). **c** E16.5 $Cx_3cr1^{GFP/+}$ embryo with surrounding YS and its dense vascular network. **d** YS tissue with CX₃CR1+ pre-macrophages at indicated time points. Pictures show a representative microscopic field of 400 × 400 µm. **e** Corresponding quantification of CX₃CR1+ cells per microscopic field of 400 × 400 µm; *** $p < 0.001$ (two-tailed Mann–Whitney test); graph shows median with interquartile range (±IQR, error bars). **f** In vivo YS staining for the endothelial marker CD31 (red) in $Cx_3cr1^{GFP/+}$ (green) embryos at E10.5. **g, h, j** Images extracted from E10.5 video sequences of CX₃CR1+ YS tissue show a cell entering the vasculature (**g**, from Supplementary Movie 1) as well as intravascular non-adhering (**h**, from Supplementary Movie 3) and rolling (**j**) cells. CX₃CR1+ cells in focus are indicated by arrowheads; direction of flow from bottom to top (**g**), left to right (**h**), left to right (**j**). **i** Quantification of intravascular CX₃CR1+ cells in the YS in an average-sized vessel; median ± IQR. **k** Velocity of intravascular non-adhering ($n = 50$) and rolling ($n = 5$) CX₃CR1+ cells in the YS at E10.5; median ± IQR. Scale bars are 1 mm (**c**), 100 µm (**d, f, g, h, j**)

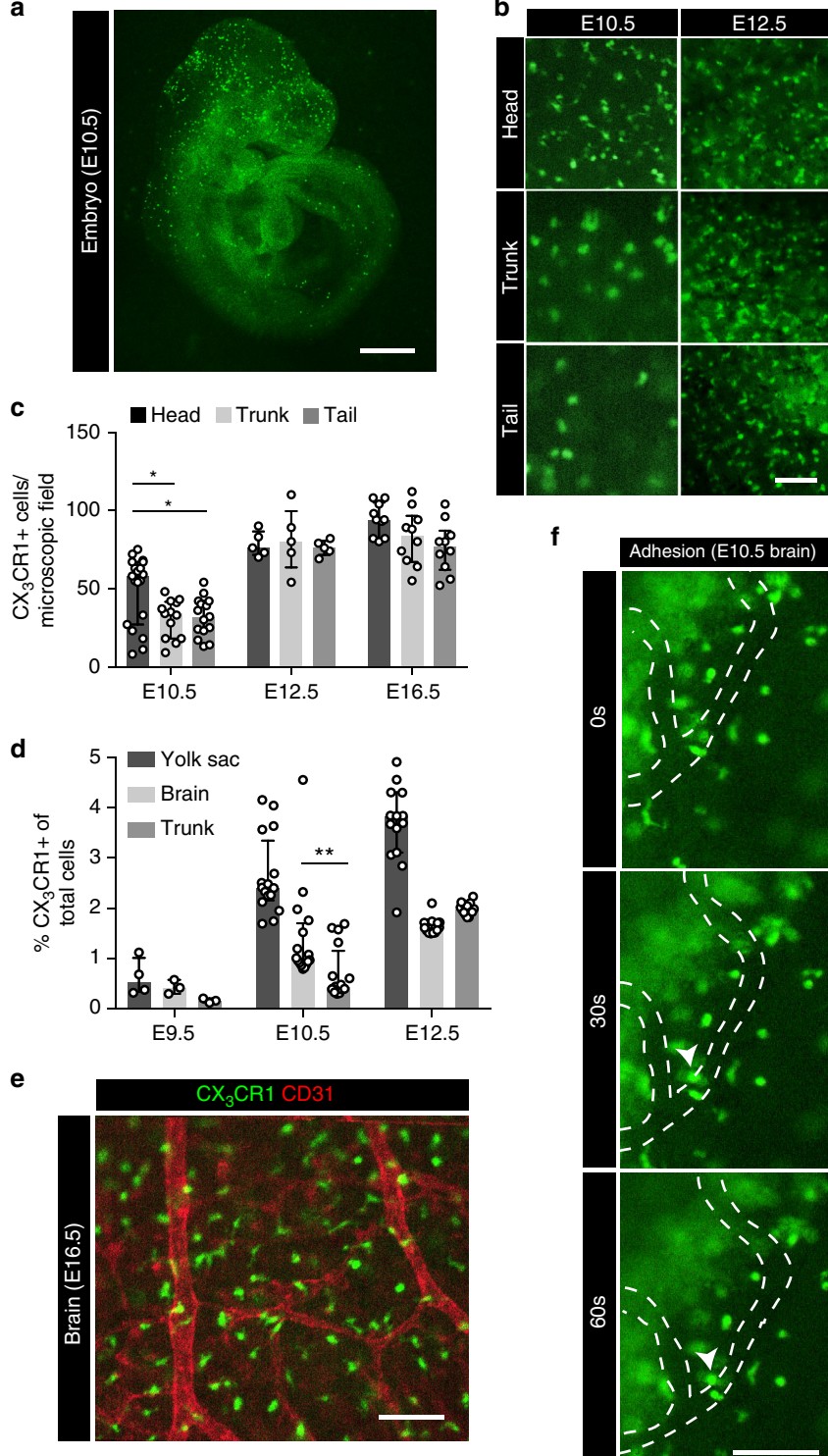

**Fig. 2** Pre-macrophages infiltrate embryonic tissues. **a** Isolated E10.5 $Cx_3cr1^{GFP/+}$ embryo. **b**, **c** Visualization (**b**) and quantification (**c**) of CX$_3$CR1+ macrophages in different embryonic regions at indicated time points; * $p < 0.05$ (one-way ANOVA with Tukey's multiple comparisons test: head vs. trunk $p = 0.0107$, head vs. tail $p = 0.0121$, trunk vs. tail $p = 0.9860$); graph shows median ± IQR. **d** Quantification by flow cytometry of CX$_3$CR1 GFP+ cells in the embryonic YS, brain and trunk at indicated time points; (**) $p = 0.0017$ (two-tailed Mann–Whitney test); median ± IQR. **e** CX$_3$CR1 GFP+ cells (green) in the embryonic brain at E16.5 additionally stained for the endothelial marker CD31 (red). **f** Image series extracted from live E10.5 video sequences (Supplementary Movie 6) show a cell attaching to the endothelium in the embryonic head region; direction of flow from top to bottom. Scale bars are 1 mm (**a**), 100 μm (**b**, **e**, **f**)

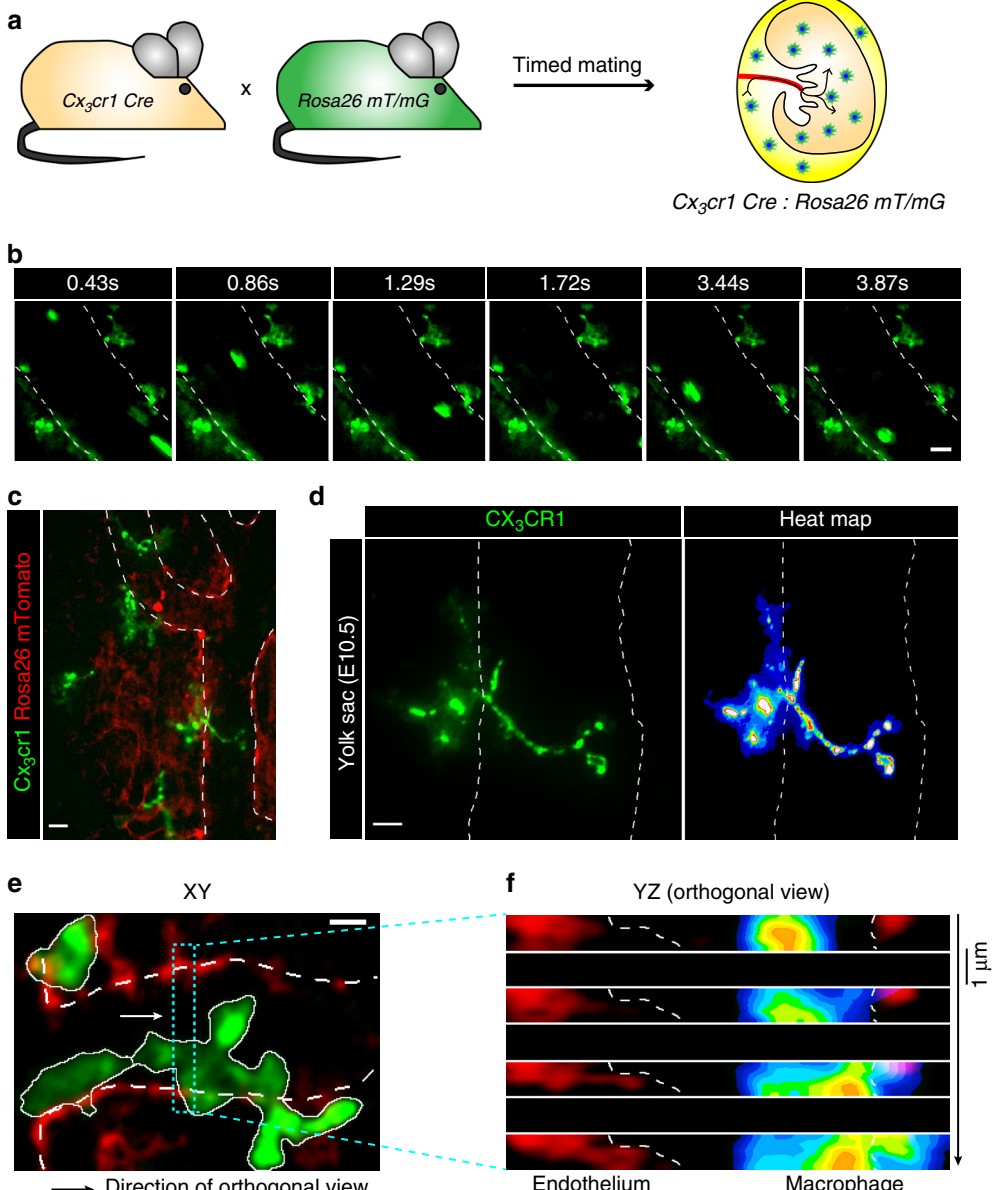

**Fig. 3** Trafficking is associated with cellular morphology. Intravital spinning disc microscopy of YS vasculature in E10.5 $Cx_3cr1^{Cre}$:$Rosa26^{mT/mG}$ embryos. **a** Schematic graph for the $Cx_3cr1^{Cre}$:$Rosa26^{mT/mG}$ mouse model. **b** Image series of CX3CR1 GFP+ (green) YS pre-macrophages; direction of flow from top to bottom (from Supplementary Movie 7). **c-f** Dendrite-shaped CX3CR1 GFP+ macrophages (green) with intravascular protrusions (**c**, **d**) further illustrated by multiplanar reconstructions in XY (**e**, from Supplementary Movie 8) and orthogonal YZ views including heat map (**f**); endothelium (mTomato, red). Scale bars are 20 μm (**b**, **c**, **e**), 10 μm (**d**)

the bloodstream was restricted to this well-defined time window of development (Fig. 6g). The difference in the number of trafficking cells at E12.5 between pulse-labeled $Csf1r^{MerCreMer}$ (Figs 5 and 6f) and $Cx_3cr1^{GFP/+}$ (Fig. 1i) mice could reflect CSF1R-labeling of EMPs as compared to matured "primitive" macrophages, the latter having completed their emigration from the YS at this time.

To dissect the different stages of macrophage development in the YS, we harnessed $Csf1r^{MerCreMer}$:$Rosa26^{tdTomato}$:$Cx_3cr1^{GFP/+}$ mice, in which we combined the E8.5 CSF1R-mediated OH-TAM pulse labeling approach with the constitutive $Cx_3cr1^{GFP/+}$ model to identify maturing macrophages. By E10.25/E10.5 a large proportion of CSF1R+ EMPs had acquired CX3CR1 (Fig. 6h, Supplementary Fig. 4c), which is in line with the recent reports on the sequence of macrophage maturation[12,21]. CSF1R+ CX3CR1+

pre-macrophages displayed a spherical shape and were frequently found to be traveling to the embryo (Fig. 6h, Supplementary Movies 3–5 and 7). F4/80 was expressed on mature macrophages with dendrites. These cells were located throughout the YS and often adjacent to YS vessels. They displayed reduced migration capacity and did not travel intraluminally (Figs. 3, 6h and Supplementary Movies 7, 8). Thus, trafficking of macrophage progenitors to the embryo correlated with defined stages of their development.

**Expansion of CSF1R⁺ macrophage progenitors.** The proliferative potential of YS EMPs has previously been demonstrated in vitro using clonogenic assays[16]. In $Csf1r^{MerCreMer}$:$Rosa26^{eYFP}$ embryos in vivo, E8.5 OH-TAM pulse-labeled CSF1R+ progenitors displayed cell clusters in tissues (Fig. 6d, e and

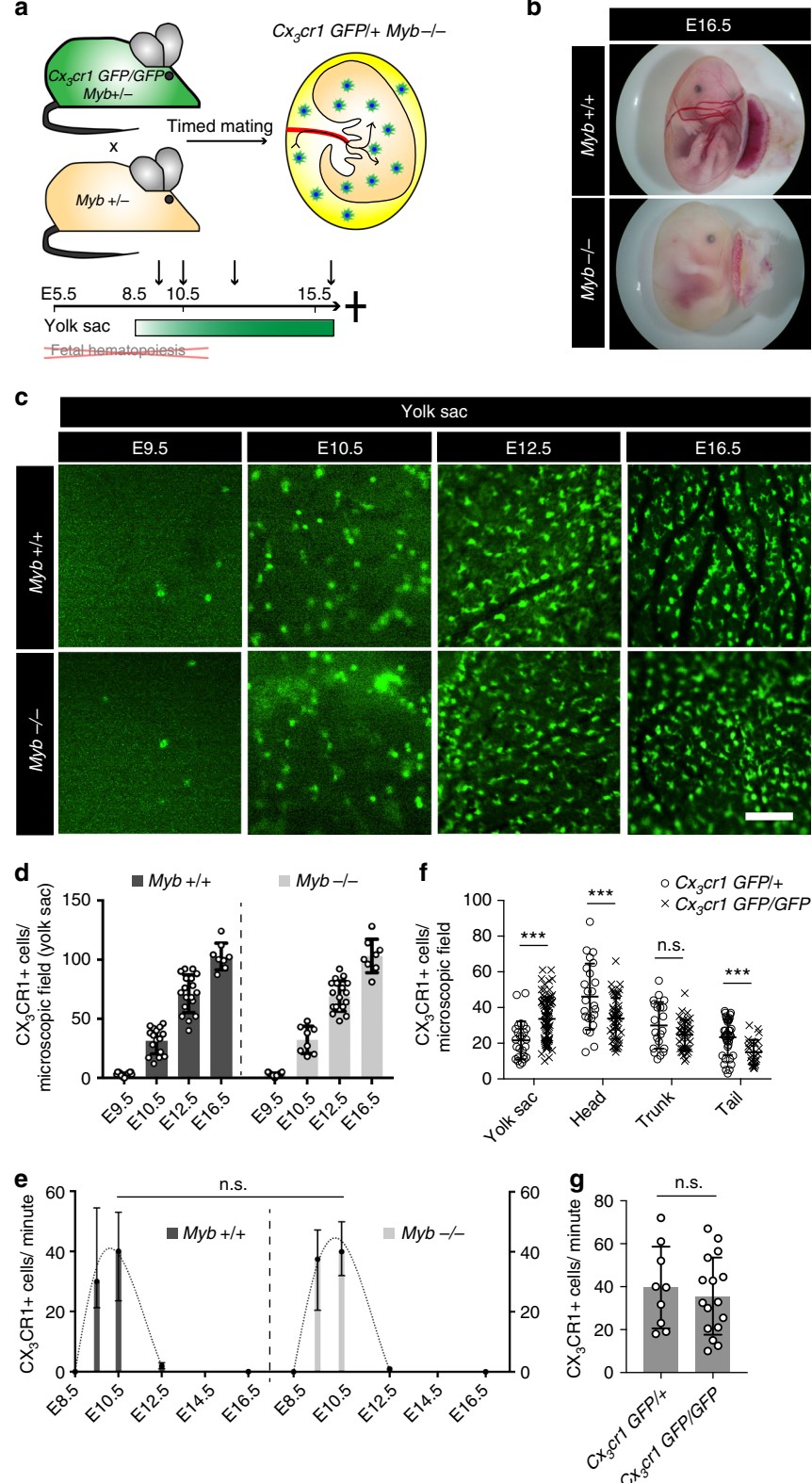

**Fig. 4** Intravascular trafficking is independent of MYB and CX₃CR1. **a** Schematic graph of the $Cx_3cr1^{GFP/+}$ $Myb^{-/-}$ mouse model. **b** Bright field images of isolated embryos at E16.5 indicating the absence of fetal definitive hematopoiesis (i.e. severe anemia). **c**, **d** Visualization (**c**) and quantification (**d**) of CX₃CR1+ cells in the YS of $Myb^{+/+}$ or $Myb^{-/-}$ mice at indicated time points; all comparisons of $Myb^{+/+}$ vs. $Myb^{-/-}$ are n.s. (two-tailed $t$-test: E9.5 $p = 0.9818$, E10.5 $p = 0.8894$, E12.5 $p = 0.6900$, E16.5 $p = 0.9543$); mean ± standard deviation (SD, error bars). **e** Quantification of intravascular CX₃CR1+ cells in a $Myb^{+/+}$ and $Myb^{-/-}$ YS on indicated time points; E10.5: n.s. (two-tailed Mann–Whitney: E10.5 $p = 0.4537$); median ± IQR. **f** Quantification of CX₃CR1+ cell densities per microscopic field of 400 × 400 µm in embryonic tissues at E10.5; *** $p < 0.001$ (two-tailed $t$-test); mean ± SD. **g** Quantification of intravascular CX₃CR1+ cells in YS of $Cx_3cr1^{GFP/+}$ and $Cx_3cr1^{GFP/GFP}$ E10.5 embryos; n.s. (two-tailed $t$-test: $p = 0.6065$); mean ± SD. Scale bar is 100 µm (**c**)

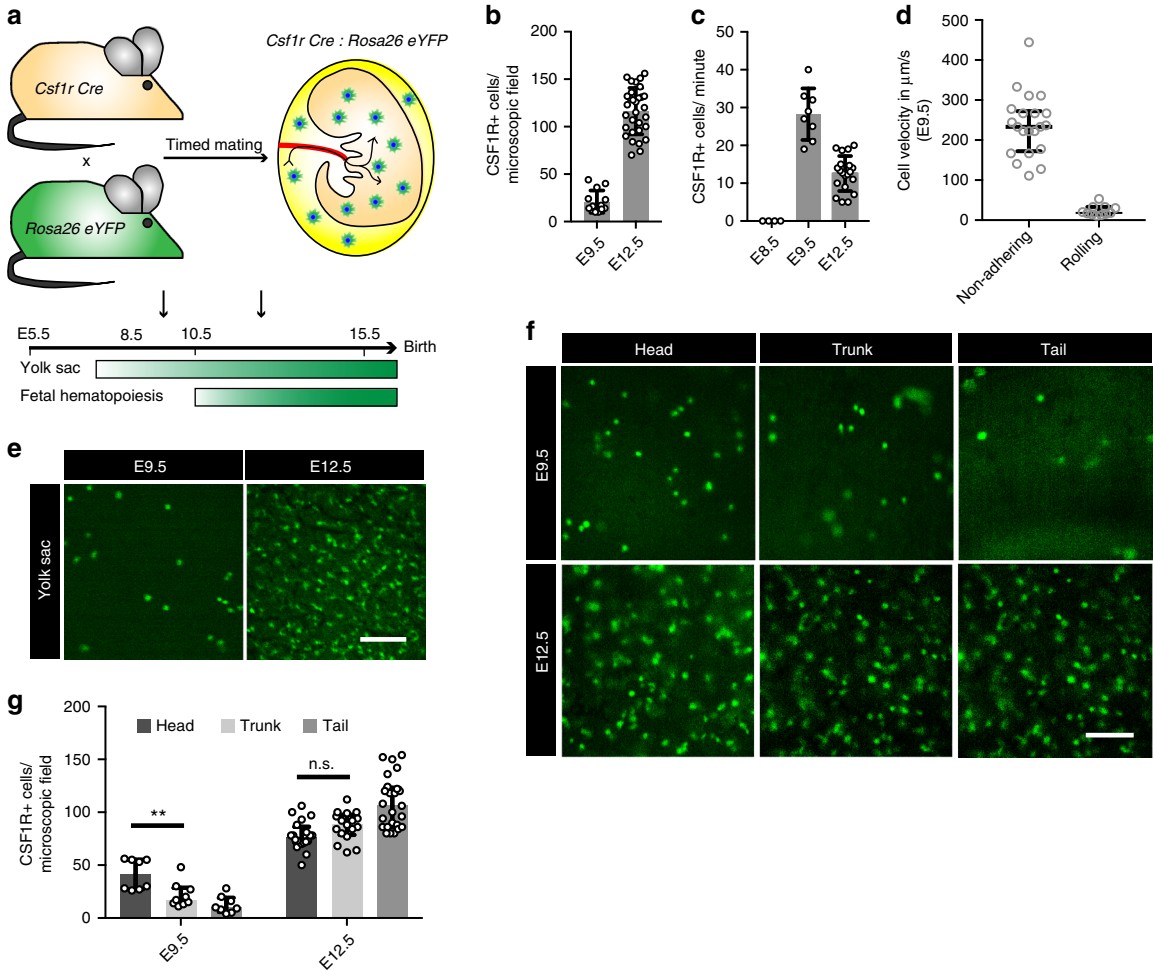

**Fig. 5** Trafficking kinetics of CSF1R+ cells are similar to pre-macrophages. **a** Schematic graph for the *Csf1r^Cre:Rosa26^eYFP* mouse model. **b**, **e** Fluorescence images of CSF1R YFP+ cells (green) in the YS (**e**) with corresponding quantifications (**b**) of cells per microscopic field at indicated time points; mean ± SD. **c** Quantification of intravascular CSF1R+ cells in an average-sized vessel; mean ± SD. **d** Velocity of intravascular non-adhering (n = 21) and rolling (n = 11) CSF1R+ cells in the YS at E9.5; median ± IQR. **f**, **g** Fluorescence images of CSF1R YFP+ cells (green) in different embryonic regions (**f**) with corresponding quantifications (**g**) of cells per microscopic field at indicated time points; ** p = 0.0072, n.s. p = 0.0569 (two-tailed Mann–Whitney test); median ± IQR. Scale bars are 100 μm (**e**, **f**)

Supplementary Movies 13, 14). In the YS, cell clusters were relatively large (about 20 cells) while in the embryo cell conglomerates mostly consisted of few YFP+ cells (2–7 cells) at E12.5 (Fig. 6e, f). This suggests that macrophage progenitors expand in the YS before their emergence and further expansion takes place inside the embryo. This is in line with the proliferation dynamics of YS-derived EMPs in the fetal liver[16] and of tissue macrophages in the epidermis of adult mice[42]. Notably, in liver sections of 1 year-old *Csf1r^MerCreMer:Rosa26^eYFP* mice pulsed at E8.5, large clusters of YFP+ Kupffer cells were present (Fig. 6i, j). Thus, the patterns of expansion initiated during early embryonic development can persist in adult mice. Future fate mapping experiments harnessing multicolor reporter mice are necessary to proof clonality of these cell clusters.

**Trafficking dynamics of KIT+ EMPs.** Application of a OH-TAM pulse at E8.5 in *Csf1r^MerCreMer:Rosa26^eYFP* embryos effectively labeled YS-derived EMPs (Supplementary Fig. 4b)[16,43]. However, maturing macrophages that arise from the first (primitive) wave of YS hematopoiesis may also be labeled once *Csf1r* has become active[6,44]. Flow cytometry analyses of the YS revealed a CD16/32 + KIT- population of primitive macrophages progenitors before

the first appearance of EMPs suggesting their independent origin in line with previous reports (Supplementary Fig. 5a)[12,23]. Only a few hours later KIT+ EMPs develop in the YS and seed the fetal liver (Supplementary Fig. 5a–d) before they differentiate into KIT- CX₃CR1+ pre-macrophages as described previously (Supplementary Fig. 5b, e)[12,23].

To more specifically determine the trafficking of YS-derived EMPs we carried out intravital microscopy in mice, in which Cre-dependent recombination was under control of the *Kit* promoter. KIT has previously been identified as a marker of EMPs[12,45], and distinguishes them from maturing YS macrophages and primitive progenitors[22,23]. In order to label KIT+ progenitors, we crossed *Kit^MerCreMer* mice with *Rosa26^mT/mG* reporter mice and applied a single dose of OH-TAM to induce GFP expression (Fig. 7a). In *Kit^MerCreMer:Rosa26^mT/mG* embryos, OH-TAM application from E7.5 to E8.5 resulted in increased labeling of EMPs (Fig. 7b–d). Intravital microscopy at E10.5 showed that numerous EMPs, or their progeny, migrated to the embryo via the YS vasculature (Fig. 7e, f and Supplementary Movie 15). Cells derived from *Kit*-expressing progenitors thereby infiltrated all embryonic regions including the brain (Fig. 7b, c, g). However, the number of trafficking cells in E8.5 pulse-labeled *Kit^MerCreMer* embryos was

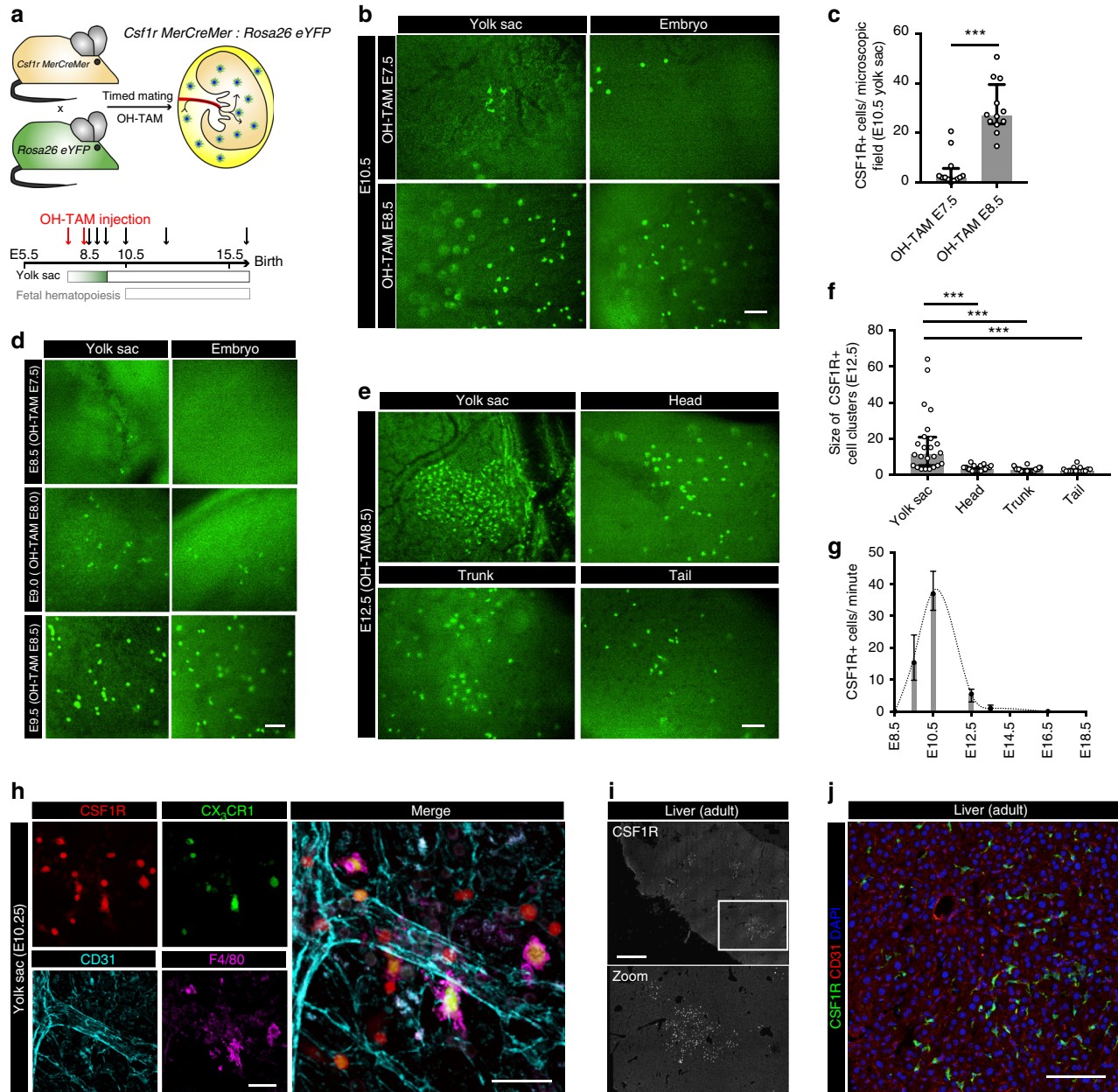

**Fig. 6** Cellular expansion and morphology of CSF1R+ progenitors. **a** Schematic graph for the $Csf1r^{MerCreMer}:Rosa26^{eYFP}$ mouse model with OH-TAM pulse labeling at E7.5 or E8.5 (red arrows) and subsequent analyses at indicated time points (black arrows). **b**, **c** Fluorescence pictures of E10.5 YS and embryo after OH-TAM pulse at E7.5 or E8.5 as indicated (**b**) with corresponding quantifications **c**; *** $p < 0.001$ (two-tailed Mann–Whitney test); median ± IQR. **d** Fluorescent pictures in the YS and embryo at E8.5, E9.0 and E9.5 with OH-TAM injection 24 h earlier. **e**, **f** Fluorescence pictures of a E12.5 YS and different embryonic regions after OH-TAM pulse labeling on E8.5 (**e**) with corresponding quantification of cluster size (**f**); *** $p < 0.001$ (Kruskal–Wallis test with Dunn's multiple comparisons test); median ± IQR. **g** Quantification of intravascular cells at indicated time points in an average-sized vessel; median ± IQR. **h** Fluorescence pictures of a E10.25 YS in a $Csf1r^{MerCreMer}:Rosa26^{tdTomato}$ (red):$Cx_3cr1^{GFP/+}$ (green) mouse model with OH-TAM pulse labeling at E8.5. Additional staining for F4/80 (purple) and CD31 (turquoise). **i**, **j** Adult liver sections of $Csf1r^{MerCreMer}:Rosa26^{eYFP}$ mice, pulse-labeled at E8.5 (**i**) with additional stainings for CD31 (red) and DAPI (blue) (**j**). Scale bars are 100 μm (**b**, **d**, **e**, **j**), 1 mm (**i**), 50 μm (**h**).

lower than in E8.5 pulse-labeled $Csf1r^{MerCreMer}$ mice at E10.5, when trafficking reached its maximum (Figs. 6g and 7e). The difference might be explained by labeling of both EMPs and differentiating "primitive" macrophages in the $Csf1r$-dependent model. However, other factors such as differences in Cre recombination efficiency might also play a role. In summary, the experiments indicate that of the different cell types labeled by applying a OH-TAM pulse, EMPs and their progeny represent an integral part trafficking from the YS to the embryo, which is in line with their role in seeding the fetal liver[13,16,46].

## Discussion

Our study provides a detailed characterization of early YS macrophage development and trafficking to embryonic tissues in high spatiotemporal resolution. Shortly after their appearance in the YS, macrophage progenitors—in both EMP and pre-macrophage stages—travel within the bloodstream to infiltrate the embryo. The trafficking process reaches a maximum around E10.5 and is mostly completed by E12.5. YS macrophages thereby traffic in spherical mobile shape. A dendrite-rich form is acquired by YS-derived macrophage progenitors once target tissues have been

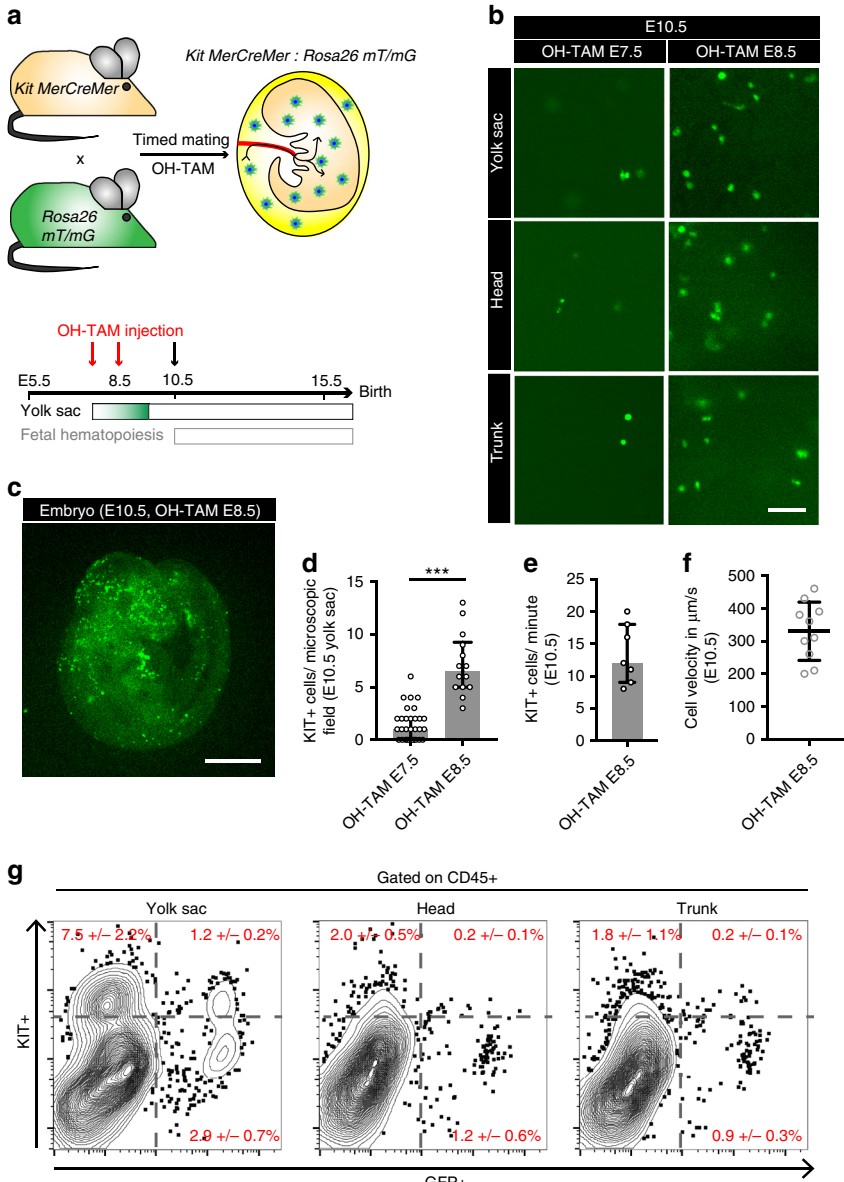

**Fig. 7** Trafficking of KIT+ EMPs. **a** Schematic graph for the *Kit^MerCreMer^:Rosa26^mT/mG^* mouse model with OH-TAM pulse labeling at E7.5 or E8.5 (red arrows) and subsequent analyses at E10.5 (black arrow). **b**–**d** Fluorescent pictures of KIT+ cells in the YS and embryo on E10.5 after pulse labeling at indicated time points (**b**, **c**) with corresponding quantifications (**d**); *** *p* < 0.001 (Mann–Whitney test); median ± IQR. **e** Quantification of labeled KIT+ cells/min in an average-sized YS vessel at E10.5; median ± IQR. **f** Velocity of intravascular non-adhering KIT+ cells in the YS at E10.5; mean ± SD. **g** Flow cytometry plots of CD45+ gated KIT vs. GFP+ cells from pulse-labeled *Kit^MerCreMer^:Rosa26^mT/mG^* mice in YS, head and trunk region at E10.5. Plots show an individual representative experiment. At least 3 embryos of minimum 2 independent litters were analyzed per time point. Scale bars are 100 μm (**b**), 1 mm (**c**)

reached and further maturation proceeds. Thus, alterations in cell morphology accompanying macrophage maturation correlate with their distribution through the bloodstream (Fig. 8). However, the transition from round cells to dendritic macrophages could be happening independently of the trafficking process within tissues.

By applying intravital microscopy we provided direct evidence that YS macrophage progenitors colonize the embryo via the vasculature. There has been an ongoing debate on how YS macrophages enter the embryo, that is, by trans-tissue migration or trans-vascular trafficking via the bloodstream. The early appearance of myeloid colony-forming units in embryonic blood before the onset of fetal definitive hematopoiesis indicates trans-vascular trafficking[13]. Further, infiltration of embryonic tissues by EMPs and maturing macrophages is impaired in mice lacking a heartbeat supporting the role of a functional circulation for macrophage trafficking[7,22,47,48,]. However, this model is limited by its dramatic impairment of overall mouse physiology resulting in early embryonic lethality[49]. In contrast to the data supporting trans-vascular migration, it has been suggested for various species that macrophages can colonize embryonic tissues independently of blood vessels via extravascular routes[25–27,35]. In line with this, functional circulation is not required for EMP emergence in the YS[22]. We interpret our data to indicate that macrophage progenitors infiltrate the embryo predominantly by intravascular trafficking. However, the microscopy tools applied in this paper

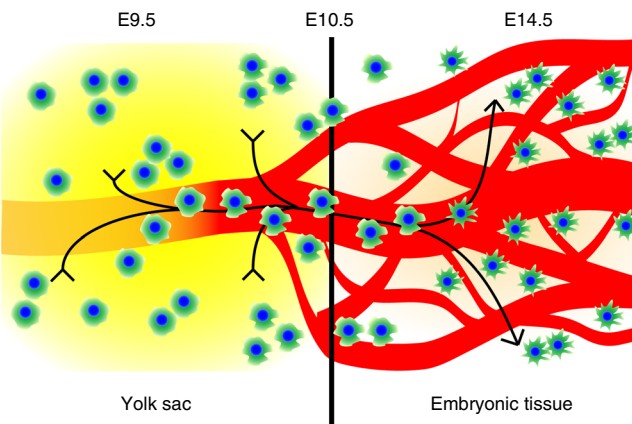

E9.5　　　　　E10.5　　　　　E14.5

Yolk sac　　　　　　　　　　Embryonic tissue

**Fig. 8** YS-derived macrophages traffic during defined stages of development. Graphical summary of morphology-dependent trafficking dynamics: macrophage progenitors expand in the YS from E8.5 onwards. They traffic to embryonic organs via vascular routes during a restricted time frame until E14.5. This trafficking period is characterized by a spherical cell shape. Along with progressive macrophage maturation an increasing amount of dendrites is formed and macrophages lose intravascular trafficking capacity

are suited for imaging fast intravital processes and, therefore, we cannot quantify nor exclude trans-tissue migration.

Macrophage trafficking reached its maximum around E10.5, and we detected hardly any intravascular CX3CR1+ pre-macrophages after E12.5. Blood flow velocity in the YS vasculature still increases at this gestational age in mice. It is half maximal at E10.5 and does not reach a plateau before E13.5[50]. Therefore, flow velocity does not seem to relevantly contribute to the trafficking of macrophages to the embryo. Further, trafficking is not influenced by the onset of fetal definitive hematopoiesis as demonstrated in *Myb*-deficient mouse embryos. The absence of CX3CR1 chemokine receptor signaling resulted in higher macrophage densities in the YS and impaired colonization of the embryo[21]; however, the quantity of intravascular cells was not altered. Thus, CX3CR1 does not affect intravascular trafficking of macrophage progenitors from the YS, it might however alter macrophage proliferation in the YS[51] or modulate survival in peripheral embryonic tissues[52,53], which remains to be determined. The molecular cues regulating macrophage trafficking from the YS to the embryo will need to be addressed in future studies. In addition to potential soluble molecules in the YS environment or cell surface receptors expressed on YS progenitors, endothelium-derived signals have recently been reported to modulate the seeding of macrophages in their destined tissue within the embryo[54].

YS hematopoiesis gives rise to multipotent cells that can develop into fetal lymphoid progenitors[55–58]. The EMPs and pre-macrophages progenitors pulse-labeled in *Csf1r*^MerCreMer^: *Rosa26*^eYFP^ embryos did not express markers of lymphoid cells. Although we did not carry out clonogenic assays and we therefore cannot formally exclude the lymphoid potential of trafficking CSF1R+ cells[22,59], our findings are in line with a recent study by McGrath and colleagues who phenotyped purified EMPs and demonstrated their lack of B-lymphoid potential[23]. Further, lineage tracing in *Rag1*^Cre^:*Rosa26*^eYFP^ mice indicated that tissue macrophages develop independently of lymphoid-primed YS progenitors[59].

Using intravital microscopy, we show that YS macrophage progenitors migrate to the embryo in both EMP and pre-macrophage stages. This is in line with the concept that EMPs

travel to the embryo to colonize the nascent fetal liver before E10.5, where they give rise to tissue macrophages and other myeloid cells (Supplementary Fig. 4b)[13,16,46]. The observation that more fluorescent cells were trafficking in E10.5 *Csf1r*^MerCreMer^ embryos than in *Kit*^MerCreMer^ mice might be explained by the labeling of both maturing macrophages derived from primitive hematopoiesis and EMPs in the CSF1R-dependent model, whereas pulse labeling in *Kit*^MerCreMer^ embryos is more specific for fate mapping of EMPs[23].

Most macrophage progenitors travel between E9.5 and E10.5; however first pre-macrophages enter the embryo around E9.0 with the onset of functional circulation. This population of macrophages predominantly infiltrates the embryonic brain (Figs. 2a–d and 5f, g and Supplementary Fig. 3a–c). It is possible that these cells arise from the first wave of KIT- CD16/32+ primitive progenitors[6]. However, this population might be heterogeneous and additional markers to unequivocally identify primitive macrophages remain to be determined. In close temporal proximity, emerging EMPs give rise to an increasing pool of macrophage progenitors that seed all embryonic tissues. Depending on the organ environment, these YS-derived resident macrophages then persist (e.g. brain, liver) or are replaced by circulating monocytes, either rapidly (e.g. intestine) or over time (e.g. lung, serous cavities)[7,16,60–62,]. In summary, our study provides novel insights into the emergence of macrophage progenitors from the YS, a key process in the establishment of the innate immune system.

## Methods

**Mice**. *Cx3cr1*^GFP/+ 37^, *Cx3cr1*^Cre 9^, *Myb*^−/− 63^, *Csf1r*^Cre^ expressing an improved Cre sequence under control of the *Csf1r* promoter[64], TAM-inducible *Csf1r*^MerCreMer 65^ and *Kit*^MerCreMer 66^, as well as *Rosa26*^eYFP 67^, *Rosa26*^mT/mG 68^ and *Rosa26*^tdTomato 69^ reporter mice have been previously described. *Csf1r*^Cre^ and *Csf1r*^MerCreMer^ were on FVB/N background, all other mice were on C57BL/6J background (backcrossed for at least 5 generations). Timed matings were carried out using 10–20 weeks old mice. Embryos were analyzed at indicated time points without determining their sex. Mice were maintained in a specific pathogen-free environment and fed standard mouse diet ad libitum. All animal procedures were performed in adherence to our project licence (55.2-1-54-2532-181-13) issued by the German regional council at the Government of Bavaria (Regierungspräsidium Oberbayern), Munich, Germany, and by the Institutional Review Board (IACUC 15-04-006) of the Memorial Sloan Kettering Cancer Center, New York, USA.

**Timed matings and pulse labeling**. Mice of desired genotypes were mated overnight. Embryonic development was estimated considering midday of the vaginal plug as embryonic day 0.5 (E0.5). Pregnant females were used for subsequent experiments on indicated time points. For genetic cell labeling, we crossed TAM-inducible *Csf1r*^MerCreMer^ or *Kit*^MerCreMer^ with *Rosa26*^eYFP^ or *Rosa26*^mT/mG^. Recombination was induced in embryos by single injection of 75 µg per g (body weight) of 4-hydroxytamoxifen (Sigma) into pregnant females at indicated time points.

**Intravital epifluorescence microscopy**. Experiments were performed in deep anesthesia. Access to the abdominal cavity of pregnant mice was provided by a cesarean section. Depending on the developmental stage, intravital imaging was predominantly carried out in separation from the mother animal (E8.5–E12.5) for reasons of image quality or in continuous connection to the maternal circulation (E12.5–E16.5). Quantity and velocity of trafficking CX3CR1+ cells in separated E10.5 embryos was similar to embryos being in continuous connection to the mother (Supplementary Fig. 1g, h). Imaging of all embryos was performed in a heated environment in 37 °C PBS with continuous temperature monitoring. Average sized vessels of the YS (Supplementary Fig. 1f) were imaged using an Olympus BX51WI epifluorescence microscope with a Plan N 10x/0.25 or PlanApo N 1.25x/0.04 objective for 3–15 min in different areas of the YS or the embryo using the Olympus cell R imaging software. Maximum total imaging periods per YS were about 60 min. Vascular borders were identified by spontaneous image contrast in the GFP channel and confirmed by an intravital endothelial staining with a rat anti-CD31 (PECAM-1) antibody (clone MEC 13.3, Biolegend, 1:100) (Figs. 1f and 2e). Cell clusters were defined as a cell-to-cell distance <2 cell diameters (Fig. 6f). After imaging, mice were sacrificed and embryos were used for further analysis.

**Intravital spinning disc confocal microscopy**. Experiments were performed in deep anesthesia. Embryos were prepared as described above for intravital epifluorescence microscopy. Analysis of YS-derived macrophage trafficking in real-time in vivo was performed in $Cx_3cr1^{Cre}:Rosa26^{mT/mG}$ embryos (E9.5, E10.5) using an upright spinning disk confocal microscope (Examiner, Zeiss, Germany) equipped with a confocal scanner unit CSU-X1 (Yokogawa Electric Corporation, Japan) and a CCD camera (Evolve, Photometrics, USA). Images were acquired with a 20x/ 1.0 NA or a 63x/ 1.0 NA water immersion objective (Plan Apochromat, Zeiss, Germany) using a laser with an excitation wavelength of 488 nm for the detection of macrophages and a laser with an excitation wavelength of 561 nm for the detection of the tomato fluorescence signal of all other tissues. Imaging and image processing was performed using Slidebook 6.0.11 Software (3i, USA) and Fiji (NIH, USA)[70].

**Antibodies for immunofluorescence**. For cryosections of mouse tissues, we used a rat anti-CD31 (PECAM-1) antibody (clone MEC 13.3, Biolegend, 1:50) and a rabbit anti-GFP antibody (polyclonal, Invitrogen Life Technologies, 1:200). Cy3- and Alexa 488-conjugated secondary antibodies (Dianova Immuno Research) were added at a dilution of 1:500. For whole mount stainings we used a rat anti-CD31-PE antibody (clone MEC13.3, Biolegend, 1:200), an armenian hamster anti-CD31 antibody (clone 2H8, Thermo Scientific, 1:500) and a rat anti-F4/80 antibody (clone: Cl:A3-1, Biorad, 1:500). Secondary antibodies used were anti-armenian hamster DyLight 405 (Jackson ImmunoResearch, 1:500) and anti-rat Alexa Fluor 647 (Invitrogen, 1:500). Nuclei were stained using Hoechst (Hoechst 33342, ThermoFisher, 1:2000) or DAPI (ThermoFisher, 1 µg/ml).

Human cryosections were stained with a primary mouse anti-CD68 antibody (clone CL1346, Sigma, 1:1000), rabbit anti-CD68 antibody (polyclonal, Sigma, 1:1000), rabbit anti-CX₃CR1 (polyclonal, Abcam, 1:100), or rat anti-EMR2 (clone W15101A, Biolegend, 1:100), as well as a secondary goat-anti-mouse IgG Cy2 antibody (Jackson ImmunoResearch, 1:100), goat anti-rat IgG Cy3 (Dianova, 1:100), goat anti-rabbit IgG Cy2 (Dianova, 1:100), or goat anti-rabbit IgG Cy3 antibody (Dako, 1:100). Nuclei were stained using Vectashield mounting medium for fluorescence with DAPI (Vector Laboratories, 1.5 µg/ml).

**Human samples and laser scanning confocal microscopy**. Preparation and analysis of human material was approved by the Institution's ethics committee (reference #337-06 amended 26/08/2013). The study protocol conformed to the ethical guidelines of the 1975 Declaration of Helsinki and written informed consent was obtained from each patient included in the study. Human samples were fixed in 4% neutral buffered formalin, dehydrated and consecutively embedded in paraffin. Blocks were sliced in 5 µm sections. Afterwards sample sections were washed and stained with indicated primary antibody for 60 min at room temperature and secondary antibodies for 30 min at room temperature. Confocal fluorescence images were obtained with a LSM880 Zeiss microscope with a 20x/0,8 dry objective and an Airyscan module. Images were acquired using ZEN software (Zeiss, Germany).

**Analysis of mouse liver sections**. Timed matings were performed as described above and $Csf1r^{MerCreMer}:Rosa26^{eYFP}$ embryos were pulse-labeled at E8.5. 1 year after birth, mice were euthanized under terminal anesthesia and livers were harvested. Organs were washed in PBS, fixed in 4% PFA and incubated in 30% sucrose overnight at 4 °C. Samples were embedded in Tissue-Tek OCT (Sakura Finetek) and cryoblocks were sliced into 15 µm sections. Sections were incubated in PBS containing BSA (5%) and normal goat IgG (1:60). Primary antibodies were added overnight at 4 °C. After washing, secondary antibodies were incubated for at least 2 h at room temperature. Nuclei were counterstained with DAPI. Sections were post-fixed with PFA 1% for 1 min and mounted with mounting medium (Vector Laboratories). Acquisitions were performed using a Leica TCS SP5 confocal microscope (Leica Microsystems) with tile scan software and 10x/0.4 (dry) as well as 20x/0.7 (water immersion) objective.

**Microscopy of mouse whole mount stainings**. YSs were mounted and visualized using a Axio Imager.M2 microscope (Zeiss, Germany) with a 20x/0.75 Plan Apochromat or 40x/0.8 Plan Neofluar objective and AxioVision Software (Zeiss, Germany) (refers to data presented in Supplementary Fig. 1d, e). Alternatively, a LSM880 Zeiss microscope with 40x/1.4 (oil) objective with ZEN black software (Zeiss, Germany) was used and data was analyzed with Imaris software (refers to data presented in Fig. 6g).

**Flow cytometry of mouse embryonic tissues**. Pregnant females were sacrificed by cervical dislocation. Embryos ranging from embryonic day (E) 8.5 to E16.5 were dissected out from the uterus and washed in 4 °C phosphate buffered saline (PBS, Invitrogen). The YS, fetal liver or embryonic regions (brain, trunk, tail) were harvested and digested in PBS containing 1 mg/ml Collagenase D (Roche), 100 U/ml Desoxyribonuclease I (Sigma) and 1% fetal bovine serum (PAA Laboratories) at 37 °C. Tissues were mechanically dissociated and passed through a 100 µm cell strainer (BD). Cells were centrifuged at 400 g for 5 min, resuspended in 4 °C PBS, plated in multi-well round-bottom plates and immunolabeled for flow cytometry

analysis. Antibody mixes were added in 1% BSA/PBS and incubated for 20 min on 4 °C.

APC-Cy7-labeled anti-CD45.2 (clone 104) antibodies were from BD Pharmingen. Brilliant violet 421-labeled anti-F4/80 (clone BM8), APC-labeled anti-B220 (clone RA3-6B2), brilliant violet 421-labeled anti-CSF1R (CD115; clone AFS98), Alexa 647-labeled anti- CX₃CR1 (clone SA011F11), PE-labeled anti-CD4 (clone H129.19), PE-labeled anti-CD19 (clone 6D5), APC-labeled anti-IL7Ra (CD127; clone A7R34), PE-labeled anti-Ter119 (clone Ter-119) and PE-labeled anti-CD16/32 (FcgRIII/II; clone 93) antibodies were from Biolegend. APC-labeled anti-KIT (CD117; clone 2B8) and PE-labeled anti-CD11b (clone M1/70) antibodies were from BD Biosciences. APC-labeled anti-Gr1 (Ly6G; clone RB6-8C5) antibodies were from eBiosciences. Flow cytometry was performed using a BD Biosciences LSR Fortessa flow cytometer and data were analyzed using FlowJo 10.

**Statistics**. Data were tested for normal distribution using the D'Agostino & Pearson omnibus test. In case of normal distribution, comparisons between groups were calculated using unpaired, two-tailed Student's t-test or one-way analysis of variance ANOVA (with Tukey's multiple comparisons test). If normal distribution was not given, Mann-Whitney test or Kruskal-Wallis test (with Dunn's multiple comparisons test) was used. A $p$-value of $<0.05$ was regarded as significant. Data is indicated as (***) $p < 0.001$, (**) $p < 0.01$, (*) $p < 0.05$, or not significant (n.s.) if $p \geq 0.05$. Data are are expressed as mean ± standard deviation (SD) if normal, or as median with interquartile range (IQR) if not distributed normally. Dots represent individual values. All graphs and calculations were generated with GraphPad Prism 7 software.

Animal experiments were analyzed in a blinded manner where possible, i.e., when the genotype of the embryos was unknown at the time of experiments (analysis of $Cx_3cr1^{GFP/+}$ vs. $Cx_3cr1^{GFP/GFP}$, and $Myb^{+/+}$ vs. $Myb^{-/-}$ mice).

**Data availability**. The authors declare that the data supporting the findings of this study are available within the article and its supplementary information files.

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

## Acknowledgements

We thank Nicole Blount and Beate Jantz for animal husbandry. We thank Sebastian Helmer, Marina Hofmann and Michael Lorenz for mouse genotyping. We thank Christina Kuhn for immunofluorescence stainings. We thank Steffen Jung, The Weizmann Institute of Science, for providing $Cx_3cr1^{Cre}$ mice. This study was supported by the SFB914, projects A01 (M.M.), A10 (C.Schu.), B01 (M.S.), B02 (S.M.), Z01 (S.M., H.I.-A.) and Z03 (R.P., C.Schei.), as well as the DZHK (German Centre for Cardiovascular Research) and the BMBF (German Ministry of Education and Research) to C.S. (High Risk High Volume Late Translational project). C.Str. is supported by a Gerok position of the SFB914. F.G. was supported by NIH/NCI P30CA008748 MSKCC core grant and NIH/NIAID 1R01AI130345-01.

## Author contributions

C.Str. and C.Schu. designed the study, performed experiments and wrote the paper; R.S. and F.W. and performed experiments; A.M and M.S. helped to optimize intravital epifluoresence imaging of the embryo and performed associated experiments; R.P. and C. Schei. provided the spinning disc confocal microscopy platform and performed associated experiments; H.C.I.-A. provided the confocal microscopy platform and performed associated experiments, E.M. performed experiments with the $Csf1r^{MerCreMer}$ :$Rosa26^{tdTomato}$:$Cx_3cr1^{GFP/+}$ mouse line; S.H. and U.J. provided human material and performed associated experiments; R.T., T.W., M.S., S.M. and F.G. discussed data and

revised the manuscript; J.F., S.Y., R.V., J.D.M., S.K. and M.M. generated and provided mice and discussed data.

## Additional information

**Competing interests:** The authors declare no competing financial interests.

