## [Peer Review File · Nature Communications]

Reviewers' comments:

Reviewer #1 (Remarks to the Author):

Comments for transmission to the authors. Trafficking of yolk sac macrophage progenitors to the embryo during defined stages of development. Stremmel et al.,

This manuscript addresses the important question of when, where, and how 'resident' macrophage precursors traffic from the yolk sac to fetal tissues during development using lineage tracing with gene promoters specific for yolk sac stage, definitive hematopoietic stage, and adult stage hematopoiesis. The authors use an advanced in vivo imaging and a novel technique for in vivo embryo isolation. While these studies show definitively the traffic patterns of these cells in early embryonic to fetal transition in mice, the traffic of pure hematopoietic stem and multipotent progenitor cells is implied, not shown in this paper.

These findings are important for the field, and deserve publication. However, at times the authors' conclusions are overstated. To this end, further clarification and experimental data would significantly improve these findings towards publication.

We suggest a considerable number of minor revisions, that we feel would enhance the understanding of this manuscript for the audience. Whilst considerable in number, we feel these minor comments would be easily addressed. In addition, we recommend a small number of experimental revisions and additions, which we feel are necessary for this manuscript to be published in Nature Communications in the near future.

Major revisions:

Comment 1 The authors should show that the *Csfr1r* reporter labelled EMPs are multi-lineage competent. Although the acronym EMP implies no lymphoid outcomes, It is important to show whether the reporter mice give rise to hematopoietic lineages of erythromyeloid as well as lymphoid types. Studies published 40 years ago demonstrated that orthotopic synchronous transplantation of yolk sac blood island cells from E8-9d donors gave rise to spleen colony forming erythromyeloid colonies as well as T cells. Recent identification and prospective isolation of several types of yolk sac hematopoietic stem cells included full lympho-myeloid reconstitution of irradiated newborn mice, and so this issue should be mentioned, and perhaps clarified.

Comment 2 In a similar vein as comment 1, is hematopoiesis in the *Cx3cr1gfp/+* mice fully functional. As the authors and others cited in the article have shown that *Cxc3cr1* is an important regulator of hematopoietic development. Therefore, the authors should demonstrate that the reporter model tracked cells do or do not miss important lineages. .

Comment 3 While the authors focus on vascular based trafficking it is hard to understand how the authors can draw conclusions about the role of para-vascular migration and its importance for macrophage trafficking from the yolk sac, as this process was not studied in the paper; these conclusions should be adjusted.

Comment 4 Whilst the authors have shown that *Cx3cr1+* pre-macrophages are not trafficked from the yolk sac at E8.5 the authors have not provided data for the trafficking of *Csfr1+* EMPs at this stage of development or indeed early time points. The authors have previously published that *Csfr1+* derived progeny infiltrate fetal tissues of the body derived from E7.5, or even as early as E6.5. It would be important if they have evidence for trafficking, or the lack of, from the earliest stages of yolk sac circulation onwards.

Comment 5 In order to show that there is no difference in kinetics between Myb^{+/+} and Myb^{-/-} it is important to provide data during the full window of trafficking E8.5 to E12.5 and especially to parallel figure 1i. Indeed, it is important to understand why the authors have addressed this question at E12.5 and show trafficking, when they claim for the rest of the manuscript that trafficking has stopped at E12.5.

Minor revisions:

Comment 6 It would be helpful if authors would distinguish between primitive and definitive yolk sac hematopoiesis, rather than lumping them together as “early embryonic origin”, as distinct from AGM-based definitive hematopoiesis. Definitive hematopoiesis has been reported to occur both in the YS and the AGM, albeit recent studies appear to favor the yolk sac quantitatively. Perhaps the authors don’t distinguish between two early sources of macrophages because their approach in this manuscript can’t distinguish them. Nevertheless, if that is the case, it should be clearly stated in the introduction and addressed in the discussion.

Comment 7 In line 88 the authors state, “We identified a restricted time frame in which these cells travel from the YS to infiltrate the embryo proper via the bloodstream and rapidly expand in numbers before becoming phenotypically mature macrophages within their tissue of residence.” This important issue could be more clearly discussed.

We note that the authors conclude a “Restricted time frame...via the bloodstream”, and as such any mention of restricted timeframe throughout the manuscript should be clear that the trafficking described is via the vasculature (line 195, 229,271).

Please provide the movie, or reference the movie, that shows the cells entering the vasculature in the yolk sac.

The authors show Cx3CR1 pre-macrophages in the bloodstream, but that does not exclude extravascular tissue-trafficking, and this should be noted.

The authors state that phenotypic maturity causes the mature pre-macrophages not to migrate but it is conceivable the transition from round cells to more dendritic cells could be happening independently from the traced yolk sac cells both in the yolk sac and in the periphery,

Comment 8 Please provide a small sentence to expand on the in vivo imaging strategy (line 96). Additionally, adjust the methods to clarify if the embryo is removed from the uterus or attached to the uterus of the live mother; the figure implies the latter.

Comment 9 In line 138 “Consistent with the trafficking of YS precursors through vascular routes, there was a short time delay of few hours between detecting pre-macrophages in the yolk sac and their subsequent appearance of macrophages in embryonic tissues.” Would there not be a short time delay by tissue-migration method also?

Please provide evidence for direct cell-to-cell contacts (line112).

Comment 10 The authors stated in Line 168 “, the primary target for YS-derived macrophages was the developing brain”, however, the manuscript did not contain any data about the kinetics and numbers of macrophages in different organs, so the statement could be clarified. In addition, in Line 170 – “macrophages in the embryonic brain could be frequently visualized in close proximity

to vascular structures" but the referenced pictures did not provide evidences for the statement. The resolution is not high enough, the pictures show random distribution of macrophages.

Comment 10 Line 174 "Taken together, these data provided direct evidence that YS pre-macrophages trafficked to the embryo proper through the bloodstream and entered respective target tissues via vascular routes." This is overstated, simultaneous establishment of vessels and pre-macrophage is not direct evidence of the mechanism of trafficking

Comment 11 We suggest that the authors rephrase "abortion material" (line 182) to human fetal material and would prefer that the specific proportion of the fetus be stated. The source of human fetal material can be stated in the methods.

Comment 12 The authors do not explain the experimental result with CD68 Figure2 in the main text. We request that the authors state that human pre-macrophages were detected with CD68 to parallel the emphasis on how mouse pre-macrophages were detected and distinguish methods between the two species.

Comment 13 The experiment referred to in line 226 did not shed any light into the molecular mechanism of trafficking because it was not rationalized on this basis. Again, the authors could clarify their thinking on what molecular mechanisms would be controlled by HSC origins and could helpfully suggest alternative hypotheses.

Comment 14 Consider revision of tense Line 234 "travelled".

Comment 15 Please state line 237 "Thus, alterations in cell morphology restrict the trafficking potential of YS macrophages." More accurately, we suggest "...correlate with restricted trafficking potential"

Comment 16 Line 238 "maturated", Consider revision to "matured"

Comment 17 The authors state that, "Cx3cr1+ pre-macrophages represented a large fraction of the Csf1r+ EMPs (Figure 6d)" Line 279. This is a qualitative statement and quantitative evaluation is needed to support this finding.

Comment 18 The authors state that "In Csf1rMer-iCre-Mer:Rosa26LSL-YFP embryos in vivo, pulse-labeled EMPs displayed a clonal expansion pattern in tissues" (Line 290 and 298). Based on the data presented, where only one color was used, one cannot draw a conclusion about the clonality. The "rainbow" mice should be used for such statements.

Comment 19 The authors state that, "In the YS, clones were relatively large (>20 cells) while in the embryo cell conglomerates mostly consisted of few Csf1r+ cells (2-7 cells)." (Lane – 292-294) This is an important observation; however, no data is presented to support it.

Comment 20 The observation of Kupffer cells as derived from Csf1r pulsed at E8.5 is important, as shown in Figure 6f and g,. Did authors observed any other blood cells or other tissues with CSFr1 derived cells in these mice?

Comment 21 The authors state that, "Thus, alterations in cell morphology accompanying macrophage maturation are less compatible with their distribution through the bloodstream." (Line 316) While the postulated causation is plausible, it is by no means proved by the available evidence and would thus best be described as a correlation.

Comment 22 Please Quantify and provide statistical evaluation (Line 333) "...and we detected hardly any intravascular Cx3cr1+ macrophages after E12.5."

Comment 23 On line 347 the authors state, "Furthermore, a significant proportion of immune cells emerge from the YS as more differentiated Cx3cr1+ pre-macrophages." – This can't be stated without quantitative analysis, which is missing.

Comment 24 The authors should rule out that vessel size affects macrophage trafficking by performing quantification on small, average and large vessels in order to see if vessel size will alter the timeframe of trafficking.

Figure Revisions:

Comment 25 Please refer the reader to the specific movies used to generate the data in the main figure legend

Comment 26 Provide statistics throughout and describe the statistical test used

Comment 27 Adjust black bars in bar graphs as they obscure data points

Comment 28 Arrowheads are far too small

Comment 29 It would be preferable to use Definitive, rather than descriptive, figure legend titles. E.g. Figure 2 and 4 legend titles should be revised to follow the style of the others.

Comment 30 Verify that the dotted white lines added to images are indeed vessels by showing co-localization of specific marker staining and state what the dotted white lines represent

Comment 31 Figure 1.

a. Figure 1 c: what is the relevance of this panel

b. Figure 1d and f: these panels show the same result

c. Figure 1 e: What is the error bar? What is the black bar representing? Illustrate the black bar in a more transparent way as it is obscuring the data points. Provide statistical evaluation

d. Figure 1g: Arrow heads are far too small throughout

e. Figure 1h. What is 30+1s label? What embryonic stage is this at?

f. Figure 1h. shows cells outside of the vessel wall and not entering

g. Figure 1j. What embryonic stage is that at?

h. Figure 1k. what is the biological significance of flowing vs rolling

i. Legend. "Line 699 (c) E16.5 Cx3cr1gfp/+ embryos with surrounding YS and its dense vascular network connected to the placenta. (d, f)." Consider revision as the YS vasculature is not connected to the placenta the YS vasculature is connected to the embryo and the embryo is connected to the placenta.

j. Legend. What is an average vessel size?

Comment 32 Figure2.

a. Figure 2 c. what statistical test was performed?

b. State what CD31 stains for

c. State what CD68 stains for

d. Gestational age is +2 weeks not +1 week.

Comment 33 Figure3.

a. Figure 3C. At what stage of development is this?

b. Figure 3d. no distinction between color of bar graphs

c. Figure 3 d,e,f,g Authors should include a statement of statistical significance and state the statistical test used for all data

d. Line 738 "Scale bar is 100µm (d). "d" should be "c"

Comment 34 Figure 5d. what stage of development?

Comment 35 Figure 6.

a. Figure 6f and g. Far too small to determine cellular resolution of staining.

b. Figure 6f and g. What is the biological significance of the images?

Comment 36 Movie S6a and 6b. Query that the scale bar is the same in each movie. The resolution of 6b is too poor to determine the result.

Comment 37 Movie S7a and 7b. Query that the scale bar is the same in each movie. The resolution of 7b is too poor to determine the result. If the scale bars are correct that the comparison is between different size vessels at each developmental age of analysis.

Reviewer #2 (Remarks to the Author):

It is now well recognized that tissue-resident macrophage populations in many adult organs are ultimately derived from hematopoiesis in the yolk sac prior to the emergence of hematopoietic stem cells (in no small part by the pioneering studies of the authors). In this paper, the authors examine the migration of macrophage precursors and hematopoietic progenitors, so-called EMP, from the yolk sac into the embryo proper.

The experiments are well described and the data are logically laid out. The authors provide convincing data, through images and multiple movies, that Cx3cr1⁺ macrophage cells that arise in the yolk sac enter the bloodstream and move into the embryo to seed various organs, starting with the head region. Use of genetically-labeled cells make these elegant tracking studies possible. Evidence that macrophage cells use the vasculature to migrate into the embryo is the main finding of this paper. While, as authors point out, the route macrophages take to colonize embryonic tissues has been controversial, recent studies of Ncx1^{-/-} mouse embryos support the role of the vasculature, particularly Ginhoux, et al. *Science* (2010), as cited by the authors, as well as Hoeffel, et al. *JEM* (2012), that shows a complete absence of macrophages in the skin of Ncx1^{-/-} embryos. Ncx1^{-/-} embryos also have normal/increased numbers of EMP in the yolk sac but lack EMP in the embryo (Lux, et al. *Blood* (2008), Frame, et al. *Stem Cells* (2016)). The normal/increased numbers of macrophage cells (this paper) and EMP in the yolk sac and the essentially complete lack of these cell types in the embryo of mouse embryos lacking a functional circulation provide strong presumptive evidence that macrophage cells and EMP seed the embryo through the circulation. The positive results presented here corroborate this conclusion with positive data.

In the introduction, the authors state that adult tissue-resident macrophage cells “develop independently of definitive hematopoiesis from EMPs in the yolk sac”. While this is true, it is also evident that CFU-Mac arise in the yolk sac before EMP and that differentiating macrophage cells are present in the yolk sac as EMP emerge, as is evident at E9.5 (Frame, et al. *Stem Cells* 2016) and at E10.25 (this paper, Figure 6d, where several Csf1R⁺, Cx3cr1⁺, F4/80⁺ cells are evident). In addition, lineage tracing studies in mice indicate that microglia arise from different yolk sac progenitors than the rest of the tissue-resident populations in the embryo. These findings indicate that the underlying model system that the authors use to interpret their data (that only EMP in the yolk sac give rise to all pre-macs) does not do justice to the complexity of hematopoiesis in the yolk sac.

The authors make use of an inducible Csf1R to mark EMP and conclude that EMP also use the circulation to enter the embryo- a conclusion consistent with studies in normal, as well as Ncx1^{-/-} embryos (as mentioned above). However, induced Csf1r-labeling at E8.5 would simultaneously mark not only some/most EMP but also the differentiating macrophage cells from the CFU-Mac that arise prior to EMP. I suspect it is these macrophage cells that are evident as Csf1R⁺, Cx3cr1⁺, F4/80⁺ cells in Figure 6d, and possibly to most of the CX3cr1⁺ pre-macrophage cells in the yolk sac, and which seed the head (Figure 2a) to generate the permanent microglia. In line with this,

the number of trafficking Csf1r+ cells per minute is nearly the same as the Cx3cr1+ pre-macrophages at E9.5 (Figures 1i and 5c). The difference at E12.5 would reflect Csf1R-labelling of EMP vs the differentiating "primitive" macrophage cells, which have migrated out of the yolk sac. These data also suggest that EMP, which peak at E10-10.5 in the yolk sac and are still trafficking at E12.5 (Fig. 5c), give rise to very few and no trafficking pre-macrophage cells at E11.5 and E12.5, respectively (Fig. 1i).

Studies conducted in Myb^{-/-} embryos, which lack hematopoietic stem cell function, confirm that the identified macrophage kinetics are independent of Myb gene expression. These latter findings are not unexpected given the pioneering studies published previously by the authors (Schulz, et al. 2012) that Myb^{-/-} embryos have normal numbers of macrophage cells in the embryo.

Minor points:

1. Is it better to show the data as bar graphs in Figure 1i?

2. Page 6, last 2 sentences of the middle paragraph: Is there a difference between "macrophage precursors" and "macrophage progenitors"?

Reviewer #1:

This manuscript addresses the important question of when, where, and how ‘resident’ macrophage precursors traffic from the yolk sac to fetal tissues during development using lineage tracing with gene promoters specific for yolk sac stage, definitive hematopoietic stage, and adult stage hematopoiesis. The authors use an advanced in vivo imaging and a novel technique for in vivo embryo isolation. While these studies show definitively the traffic patterns of these cells in early embryonic to fetal transition in mice, the traffic of pure hematopoietic stem and multipotent progenitor cells is implied, not shown in this paper.

These findings are important for the field, and deserve publication. However, at times the authors’ conclusions are overstated. To this end, further clarification and experimental data would significantly improve these findings towards publication.

We suggest a considerable number of minor revisions, that we feel would enhance the understanding of this manuscript for the audience. Whilst considerable in number, we feel these minor comments would be easily addressed. In addition, we recommend a small number of experimental revisions and additions, which we feel are necessary for this manuscript to be published in Nature Communications in the near future.

We thank the reviewer for these comments. We agree that this manuscript addresses an important biological question and that the established model of intravital microscopy allows for studying the trafficking patterns of macrophage progenitors from the yolk sac (YS) to the embryo proper.

In the revised manuscript, we have addressed all issues raised by this reviewer by making substantial changes in the manuscript text and by carrying out several new experiments. These include flow cytometry analysis of the hematopoietic lineages labeled in our mouse models and establishing a fate mapping analysis in *Kit*^{MerCreMer} embryos to specifically address the *in vivo* trafficking of erythro-myeloid progenitors (EMPs). Our response to your comments, which have guided us in improving this manuscript, are appended below.

Major revisions:

Comment 1 The authors should show that the Csf1r reporter labelled EMPs are multi-lineage competent. Although the acronym EMP implies no lymphoid outcomes, It is important to show whether the reporter mice give rise to hematopoietic lineages of erythromyeloid as well as lymphoid types. Studies published 40 years ago demonstrated that orthotopic synchronic transplantation of yolk sac blood island cells from E8-9d donors gave rise to spleen colony forming erythromyeloid colonies as well as T cells. Recent identification and prospective isolation of several types of yolk sac hematopoietic stem cells included full lympho-myeloid reconstitution of irradiated newborn mice, and so this issue should be mentioned, and perhaps clarified.

We thank the reviewer for this important comment. We agree that there is some controversy in the field regarding YS lymphoid potential.

We are aware of the extensive work by Moore and Owen 40 years ago, suggesting that all haemopoietic stem cells are formed in the early YS^{1, 2}. However, this work was challenged soon afterwards by works of Lassila, Dieterlen-Lievre, Belo and others. Using quail-chick as well as chick-chick YS-embryo chimeras, they demonstrated that lymphoid stem cells in the chicken do not originate primarily in the YS³, but from an intraembryonic source^{4, 5}.

First reports regarding YS lymphoid potential in mice also date back for decades to work of the Auerbach group^{6, 7}.

Importantly, Palis and colleagues recently published an in-depth analysis of subfractions of YS progenitors and thereby demonstrated that YS B-lymphoid potential is confined to non-EMP fractions while erythromyeloid progenitors lack B cell potential⁸. Furthermore, they elaborated that YS EMPs have multilineage potential and give rise to erythrocytes, macrophages and neutrophils⁸. These lineages are also labelled *in vivo* by Tamoxifen pulse labeling of E8.5 *Csf1r*^{MerCreMer}; *Rosa26*^{eYFP} embryos, as demonstrated previously by our group. YFP expression is thereby induced in YS *Csf1r*+ EMPs and their progeny resulting in labeling of macrophages but also of erythrocytes and neutrophils⁹.

To address the reviewer's comments we performed flow cytometry analysis using this *Csf1r*-dependent fate mapping model. We show that EMPs differentiated to erythrocytes, macrophages and neutrophils (Supp. Fig. 4b), while lymphoid cells were not labelled (Supp. Fig. 4d). This indicates that our analysis was not biased by labeling of potential YS lymphoid cells. Our findings are in line with the work of McGrath and also with the recent fate mapping analysis of lymphoid progenitors using *Rag1*^{Cre} mice, in which tissue macrophage were only minimally labeled indicating that they emerge independently of YS lymphoid progenitors¹⁰.

In addition to including this new experimental data in the supplementary material, we now address this issue in the discussion section. The paragraph reads as follows: "*The EMPs and pre-macrophages pulse-labeled in Csf1r^{MerCreMer}; Rosa26^{eYFP} embryos did not express markers of lymphoid cells. Although we did not carry out clonogenic assays and we therefore cannot formally exclude the lymphoid potential of trafficking Csf1r+ cells^{10, 11}, our findings are in line with a recent study by McGrath and colleagues who phenotyped purified EMPs and demonstrated their lack of B-lymphoid potential⁸. Further, lineage tracing in Rag1^{Cre}; Rosa26^{eYFP} mice indicated that tissue macrophages develop independently of lymphoid-primed YS progenitors¹⁰*".

Supplementary Figure 4: *Csf1r* pulse labeling model tracks EMPs. (a) Schematic graph for the *Csf1r^{MerCreMer};Rosa26^{eYFP}* mouse model with TAM pulse labeling. (b) Analysis of labeling efficiency by FACS in the E12.5 fetal liver for the erythroid and myeloid lineage as indicated. (c) FACS analysis of *Cx3cr1* expression in *Csf1r MerCreMer : Rosa26 eYFP+* cells in the yolk sac at E12.5. (d) FACS screening for the expression of indicated lymphoid markers in the *Csf1r^{MerCreMer};Rosa26^{eYFP}*+ pulse-labeled cell population at E12.5 in the yolk sac and fetal liver. FACS plots show an individual representative experiment. At least 3 embryos of minimum 2 independent litters were analyzed per time point.

Comment 2 In a similar vein as comment 1, is hematopoiesis in the *Cx3cr1gfp/+* mice fully functional. As the authors and others cited in the article have shown that *Cx3cr1* is an important regulator of hematopoietic development. Therefore, the authors should demonstrate that the reporter model tracked cells do or do not miss important lineages.

We thank the reviewer for this comment. We now include flow cytometry analysis of haematopoietic cells of *Cx3cr1 gfp/+* mice carried out at different timepoints of embryonic development.

1) To address the question whether hematopoiesis is functional in *Cx3cr1 gfp/+* embryos, we crossed heterozygous *Cx3cr1 gfp/+* mice and analyzed their embryos at E16.5. We found that the proportion of neutrophils, monocytes/macrophages, erythrocytes and B-lymphocytes among fetal hematopoietic cells was not different to littermate controls, i.e. wildtype (*Cx3cr1 +/+*) and functional knockout (*Cx3cr1gfp/gfp*) embryos. This indicates that hematopoiesis was not altered by the knock-in of GFP in one or both alleles. The information was added in the data supplement (Supp. Fig. 3d-f).

2) Secondly, we analyzed hematopoietic cells in the fetal liver and yolk sac of E12.5 *Cx3cr1 gfp/+* embryos to further define the tracked cells. We show that the *Cx3cr1 gfp/+* reporter model labels macrophage progenitors. Erythrocytes, neutrophils and lymphocytes were not labelled (Supp. Fig. 1b, c). Thus, in embryonic development *Cx3cr1 gfp/+* mice represent a suitable model to study the trafficking of pre-macrophages and macrophages.

As a side note, in adult mice *Cx3cr1* does not represent a specific marker for macrophages: I. In addition to macrophages, it is expressed on blood monocytes, dendritic cells and natural killer cells¹². II. Its expression is lost on some tissue macrophages, eg. liver Kupffer cells, after birth¹³.

Supplementary Figure 1: The *Cx3cr1^{gfp/+}* reporter mouse model effectively labels pre-macrophages in the yolk sac. (a) Schematic graph of the *Cx3cr1^{gfp/+}* mouse model used for embryonic analyses. (b, c) FACS analysis of fetal liver and yolk sac cells at E12.5 as indicated. (b) Quantification of Cx3cr1 GFP+ cells in the erythroid (Ter119+) and myeloid lineage; CD45+ F4/80+ Gr1- macrophages (population I) and CD45+ F4/80- Gr1+ neutrophils (population II). (c) Surface marker profile of CD45+ Cx3cr1 GFP+ cells screened for Kit, CD16/32, F4/80 and CD11b (myeloid markers) as well as B220, CD19, CD127 and CD4 (lymphoid markers). FACS plots (b, c) show an individual representative experiment. At least 3 embryos of minimum 2 independent litters were analyzed per time point. (d, e) YS whole mount staining of a *Cx3cr1^{gfp/+}* YS (green) at E10.5 (d) and E8.5 (e). Endothelium was stained by CD31 (red). Nuclei were visualized by Hoechst (d). (f) Quantification of YS vessel size used for trafficking quantifications at indicated time points. (g, h) Quantification of trafficking kinetics in connection to maternal circulation at E10.5. (g) Quantification of intravascular Cx3cr1+ cells in an average sized vessel. (h) Velocity of flowing Cx3cr1+ cells in the yolk sac vasculature. Scale bars are 100 μm (d, e).

Supplementary Figure 3: Development and distribution of Cx3cr1⁺ cells in the absence of *Myb* and Cx3cr1. (a-c) Quantification of Cx3cr1⁺ cells per microscopic field of 400 μm x 400 μm in *Myb*^{-/-} and *Myb*^{+/+} mice in indicated embryonic regions at indicated time points. (d-f) Quantification of different cell lines by indicated markers in the E16.5 fetal liver of *Cx3cr1*^{gfp/gfp}, *Cx3cr1*^{gfp/+} and *Cx3cr1*^{+/+} mice. In all individual panels, comparisons between different genotypes did not show significant differences (Mann-Whitney test).

Comment 3 While the authors focus on vascular based trafficking it is hard to understand how the authors can draw conclusions about the role of para-vascular migration and its importance for macrophage trafficking from the yolk sac, as this process was not studied in the paper; these conclusions should be adjusted.

We agree with the reviewer that this study focused on vascular based migration. Considering the vast number of macrophage progenitors trafficking per minute in the YS vessels imaged, we believe that it is not overstated that intravascular migration is the predominant way of trafficking to the embryo. Indeed, para-vascular migration cannot be excluded. We therefore adjusted the conclusions in the discussion section as follows: “*Our data indicate that intravascular trafficking is the predominant way of macrophage progenitors infiltrating the embryo proper. However, the microscopy tools applied in this paper are suited for imaging fast intravital processes and, therefore, we cannot quantify nor exclude trans-tissue migration*”.

Comment 4 Whilst the authors have shown that Cx3cr1+ pre-macrophages are not trafficked from the yolk sac at E8.5 the authors have not provided data for the trafficking of Csfr1+ EMPs at this stage of development or indeed early time points. The authors have previously published that Csfr1+ derived progeny infiltrate fetal tissues of the body derived from E7.5, or even as early as E6.5. It would be important if they have evidence for trafficking, or the lack of, from the earliest stages of yolk sac circulation onwards.

We thank the reviewer for this comment. To address this issue, we reinvestigated the development and trafficking of Csfr1+ and Kit+ cells in the YS by different experimental approaches, namely by pulse labelling of *Csfr1*^{MerCreMer} and *Kit*^{MerCreMer} embryos at days E7.5 and E8.5.

1) Pulse labelling with OH-TAM at E7.5 marked only a very small population of Csfr1+ cells in the YS of *Csfr1*^{MerCreMer};*Rosa26*^{eYFP} embryos. The number of labeled cells strongly increased when injections were carried out at E8.5 (Fig. 6b, c). As discussed in the revised manuscript, the Csfr1-dependent model is expected to label both maturing macrophages of primitive hematopoiesis and EMPs. To more specifically address the trafficking of EMPs, we harnessed Kit expression in YS EMPs⁸, and carried out additional pulse labeling experiments in *Kit*^{MerCreMer} embryos. In a similar vein to the experiments with Csfr1 embryos, pulse labelling with OH-TAM at E7.5 marked only a small population of Kit+ cells in the YS, whereas labeling increased following injections at E8.5 (Fig. 7b, d). These experiments indicate that first macrophage progenitors appear from E7.5 onwards, however, substantial labelling is only achieved from embryonic day E8.5.

2) We performed additional intravital microscopy experiments and analyzed the early trafficking phase from E8.5 to E9.5 regarding the appearance of YFP+ Csfr1 pulse-labeled cells in the YS and embryo proper (Fig. 6d). While YFP+ Csfr1 pulse-labeled cells were present in the YS at E8.5, they appeared in the embryo slightly later around E9.0 (Fig. 6d). This finding goes in line with previous studies showing the first appearance of Csfr1+ cells in the yolk sac around E8 and in the embryo proper around E9^{14, 15}. Moreover, it provides indirect evidence for an onset of intravascular trafficking between E8.5 and E9.0.

3) We tested our constitutive *Csfr1*^{Cre};*Rosa26*^{eYFP} mouse model and did not detect any fluorescent intravascular cells at E8.5 (Fig. 5c). This is in line with previous studies reporting the onset of circulation and the progressive formation of a functional vascular network in the YS between E8.25 and E9.25 respectively^{16, 17, 18}. Other studies concluded from indirect observations that EMP trafficking starts around E9.25^{11, 16}, Fig. 6d, g), which corresponds to our in vivo microscopy data (Fig. 5c, 6g).

4) We would like to clarify the reviewer's statement that "*The authors have previously published that $Csfr1+$ derived progeny infiltrate fetal tissues of the body derived from E7.5, or even as early as E6.5*".

We reported earlier on the fate mapping of YS progenitors by pulse labeling $Csfr1^{MerCreMer};Rosa26^{eYFP}$ embryos at E8.5, which marked YS $Csfr1+$ derived cells and their progeny and allowed their study following infiltration of fetal tissues¹⁹. In a follow-up paper to this project we showed that this pulse labeling model marks YS EMPs and thus lineages of both myeloid and erythroid cells, however, only tissue macrophages were not continuously replaced and persisted over time⁹. In the latter paper, we analyzed $Csfr1^{Cre};Rosa26^{eYFP}$ embryos, in which the Cre recombinase, under control of the $Csfr1$ promoter, was constitutively active. In this model, $Csfr1+$ fluorescent cells became visible from E8.5 (Fig. 1b of the paper by Gomez-Perdiguero et al.⁹). In the same line, pulse-labeling of $Csfr1^{MerCreMer};Rosa26^{eYFP}$ embryos at E6.5 did not induce labeling in YS cells (Suppl. Fig. 2b of the paper by Gomez-Perdiguero et al.⁹). Together, these experiments demonstrated that $Csfr1+$ cells and their progeny appear from E8.25 (in $Csfr1^{Cre}$ mice). These earlier observations support our present findings that labeling can be achieved by OH-TAM application in $Csfr1^{MerCreMer}$ mice from E7.5, however, quantitative labeling is only achieved from E8.5.

Figure 5: Trafficking kinetics of $Csfr1+$ cells are similar to pre-macrophages. (c) Quantification of intravascular $Csfr1+$ cells in an average-sized vessel.

Figure 6: Cellular expansion and morphologic changes in pulse-labeled *Csflr*⁺ progenitors. (a) Schematic graph for the *Csflr*^{MerCreMer};*Rosa26*^{eYFP} mouse model with TAM pulse labeling at E7.5 or E8.5 (red arrows) and subsequent analyses at indicated time points (black arrows). (b, c) Fluorescence pictures of E10.5 yolk sac and embryo proper after TAM pulse at E7.5 or E8.5 as indicated (b) with corresponding quantifications (c); (***) $p < 0.001$ (Mann-Whitney test). (d) Fluorescent pictures in the YS and embryo proper at E8.5, E9.0 and E9.5 with TAM injection 24h earlier. (e, f) Fluorescent pictures of a E12.5 YS and different embryonic regions after TAM pulse labeling on E8.5 (e) with corresponding quantification of cluster size (f); (***) $p < 0.001$ (Kruskal-Wallis test). (g) Quantification of intravascular cells at indicated time points in an average-sized vessel. (h) Fluorescence pictures of a E10.25 yolk sac in a *Csflr*^{MerCreMer};*Rosa26*^{tdTomato} (red);*Cx3cr1*^{gfp/+} (green) mouse model with TAM pulse labeling at E8.5. Additional staining for F4/80 (purple) and CD31 (turquoise). (i, j) Adult liver sections of *Csflr*^{MerCreMer};*Rosa26*^{eYFP} mice, pulse-labeled at E8.5 (i) with additional stainings for CD31 (red) and DAPI (blue) (j). Scale bars are 100 μm (b, d, e, j), 1 mm (i), 50 μm (h).

Figure 7: Trafficking of Kit⁺ EMPs. (a) Schematic graph for the *Kit^{MerCreMer}:Rosa26^{mT/mG}* mouse model with TAM pulse labeling at E7.5 or E8.5 (red arrows) and subsequent analyses at E10.5 (black arrow). (b, c, d) Fluorescent pictures of Kit⁺ cells in the yolk sac and embryo proper on E10.5 after pulse-labeling at indicated time points (b, c) with corresponding quantifications (d); (***) $p < 0.001$ (Mann-Whitney test). (e) Quantification of labeled Kit⁺ cells/ minute in an average-sized YS vessel at E10.5. (f) Velocity of intravascular non-

adhering Kit⁺ cells in the YS at E10.5. (g) FACS plots of CD45⁺ gated Kit⁺ versus pulse-labeled Kit^{MerCreMer}:Rosa26^{mT/mG}⁺ cells in YS, head and trunk region at E10.5. FACS plots show an individual representative experiment. At least 3 embryos of minimum 2 independent litters were analyzed per time point. Scale bars are 100 μ m (b), 1 mm (c).

Comment 5 In order to show that there is no difference in kinetics between Myb^{+/+} and Myb^{-/-} it is important to provide data during the full window of trafficking E8.5 to E12.5 and especially to parallel figure 1i. Indeed, it is important to understand why the authors have addressed this question at E12.5 and show trafficking, when they claim for the rest of the manuscript that trafficking has stopped at E12.5.

We thank the reviewer for raising this point.

1) To address the first comment, we performed additional experiments and analyzed the trafficking of Cx3cr1 gfp⁺ cells in Myb-deficient and wildtype (Myb^{+/+}) littermates during the E8.5-E12.5 time window. Thereby, we found no differences in the kinetics of YS macrophage trafficking between Myb^{+/+} and Myb^{-/-} (Fig. 4e).

2) In the second comment, the reviewer asks “*why the authors have addressed this question at E12.5 and show trafficking, when they claim for the rest of the manuscript that trafficking has stopped at E12.5*”.

Trafficking of macrophages and their progenitors to the embryo proper via the vasculature peaked at E10.5, followed by a rapid decrease (>90%) in the number of intravascular migrating cells. At E12.5 a very low number of trafficking cells (2-3 cells/minute) was still present, whereas from E14.5 intravascular cells were not detected anymore. In the revised manuscript we have corrected this issue. The idea to investigate E12.5 resulted from the concept that the onset of quantitatively relevant definitive hematopoiesis in the fetal liver potentially influences the trafficking dynamics of hematopoietic cells from the yolk sac to the embryo proper. Therefore, we used the myb mouse model to investigate potential differences in the presence and absence of definitive myb-dependent hematopoiesis. However, we now provide the analysis of macrophage trafficking during the full window between E8.5 and E12.5, in which we did not detect any differences between wildtype embryos and myb knockouts (Fig. 4e).

Figure 4: Intravascular trafficking of pre-macrophages is independent of Myb and Cx3cr1. (e) Quantification of intravascular Cx3cr1⁺ cells in a Myb^{+/+} and Myb^{-/-} yolk sac on indicated time points; n.s. (t-test).

Minor revisions:

Comment 6 It would be helpful if authors would distinguish between primitive and definitive yolk sac hematopoiesis, rather than lumping them together as “early embryonic origin”, as distinct from AGM-based definitive hematopoiesis. Definitive hematopoiesis has been reported to occur both in the YS and the AGM, albeit recent studies appear to favor the yolk sac quantitatively. Perhaps the authors don’t distinguish between two early sources of macrophages because their approach in this manuscript can’t distinguish them. Nevertheless, if that is the case, it should be clearly stated in the introduction and addressed in the discussion.

We thank the reviewer for this important comment.

1) As suggested, we have revised the manuscript text in order to more appropriately describe the different stages of early hematopoiesis.

The revised paragraph in the introduction section now reads as follows “*Macrophages have been proposed to develop in the YS in two successive waves: First, primitive macrophage progenitors appear around E7.25 in the YS and predominantly infiltrate the brain after the onset of circulation^{15, 20, 21, 22}. Second, multipotent erythro-myeloid progenitors (EMPs) develop from E8.25 onwards and represent a major source of tissue resident macrophages^{9, 23, 24}. Significantly later, starting from E10.5, hematopoietic stem cells (HSCs) arise in the aorto-gonado-mesonephros (AGM) region and migrate to the liver to initiate fetal definitive hematopoiesis which later shifts to the BM^{25, 26, 27}. In adult life, monocytes and other short-lived myeloid cells are continuously replaced, while tissue macrophages can self-renew and persist independently of definitive BM hematopoiesis for prolonged periods of time, potentially throughout life^{9, 19, 28}”.*

2) We also addressed this comment by performing new experiments. In this manuscript we now provide evidence by Tamoxifen application in *Csf1r^{MerCreMer}:Rosa26^{eYFP}* embryos that YS-derived EMPs were efficiently labeled (Supplementary Fig. 4b)^{9, 29}. However, maturing macrophages that arise from the first (primitive) wave of YS hematopoiesis may also be labeled once *Csf1r* has become active^{24, 30}. We therefore agree with the reviewer that the distinction between primitive and definitive YS hematopoiesis was not clear.

To address this issue, we harnessed *Kit* expression in YS EMPs⁸ and carried out additional pulse labeling experiments in *Kit^{MerCreMer}* embryos³¹. This enabled us to visualize the trafficking of *Kit*⁺ EMPs and their progeny, and complements the fate mapping analysis in *Csf1r^{MerCreMer}* embryos.

The new data was added in figure 7. The new results section reads as follows: “*To more specifically determine the trafficking of YS-derived EMPs we carried out intravital microscopy in mice, in which Cre-dependent recombination was under control of the Kit promoter. Kit has previously been identified as a marker of EMPs^{20, 32}, and distinguishes them from maturing YS macrophages and primitive progenitors^{8, 11}. In order to label Kit+ progenitors, we crossed Kit^{MerCreMer} mice with Rosa26^{mT/mG} reporter mice and applied a single dose of OH-TAM to induce GFP expression (Fig. 7a). In Kit^{MerCreMer}:Rosa26^{mT/mG} embryos, OH-TAM application from E7.5 to E8.5 resulted in increased labeling of EMPs (Fig. 7b-d). Intravital microscopy at E10.5 showed that numerous EMPs, or their progeny, migrated to the embryo proper via the YS vasculature (Fig. 7e, f and Supplementary Movie 9). Kit+ cells thereby infiltrated all embryonic regions including the brain (Fig. 7b, c and g). However, the number of trafficking cells in E8.5 pulse-labeled Kit^{MerCreMer} embryos was lower than in E8.5 pulse-labeled Csf1r^{MerCreMer} mice at E10.5, when trafficking reached its maximum (Fig. 7e, Fig. 6g). The difference might be explained by labeling of both EMPs and differentiating “primitive” macrophages in the Csf1r-dependent model. In summary, the experiments*

indicate that of the different cell types labeled by applying a OH-TAM pulse, EMPs and their progeny represent an integral part trafficking from the YS to the embryo proper, which is in line with their role in seeding the fetal liver^{9, 21, 33}.

Comment 7 In line 88 the authors state, “We identified a restricted time frame in which these cells travel from the YS to infiltrate the embryo proper via the bloodstream and rapidly expand in numbers before becoming phenotypically mature macrophages within their tissue of residence.” This important issue could be more clearly discussed.

We thank the reviewer for this comment. In this one-sentence summary cited from the end of the introduction section, we suggest a sequence of events which indeed is not as clear when considering the experimental data provided. Specifically, the process of macrophage trafficking but not their expansion was characterized in detail. We therefore rephrased this sentence as follows: *“We identified a restricted time frame in which these cells travel from the YS to infiltrate the embryo proper via the bloodstream before becoming phenotypically mature macrophages within their tissue of residence.”*

In the revised manuscript, the meaning of this sentence is discussed more clearly in various parts of the results section as well as in the discussion. For example, the first section in the revised discussion reads as follows: *”... Shortly after their appearance in the YS, macrophage progenitors - in both EMP and pre-macrophage stages - travel within the bloodstream to infiltrate the embryo proper. The trafficking process reaches a maximum around E10.5 and is mostly completed by E12.5. YS macrophages thereby traffic in spherical mobile shape. A dendrite-rich form is acquired by YS-derived macrophage progenitors once target tissues have been reached and further maturation proceeds. ...”*

We note that the authors conclude a “Restricted time frame...via the bloodstream”, and as such any mention of restricted timeframe throughout the manuscript should be clear that the trafficking described is via the vasculature (line 195, 229,271).

In the revised manuscript we now link the “time window of macrophage trafficking” to “intravascular migration” more clearly.

Please provide the movie, or reference the movie, that shows the cells entering the vasculature in the yolk sac.

The respective movie has been included in the manuscript (Supp. Movie 1).

The authors show Cx3CR1 pre-macrophages in the bloodstream, but that does not exclude extravascular tissue-trafficking, and this should be noted.

We agree with the reviewer that extravascular tissue trafficking cannot be excluded. We have added the following statement in the discussion section: *“... the microscopy tools applied in this paper are suited for imaging fast intravital processes and, therefore, we cannot quantify nor exclude trans-tissue migration”.*

The authors state that phenotypic maturity causes the mature pre-macrophages not to migrate but it is conceivable the transition from round cells to more dendritic cells could be happening independently from the traced yolk sac cells both in the yolk sac and in the periphery.

It is correct that we do not have proof for a direct link between morphology and trafficking behavior. We also agree with the reviewer that the transition from round cells to more dendritic cells could be happening independently. However, using high resolution microscopy we made the observation that pre-macrophages only travelled as round-shaped cells. In contrast, macrophages with dendrites were only found within YS and embryonic tissues. Therefore, our data suggest an association between cell shape and trafficking behavior.

In the revised manuscript we now state this correlation in the results section. The text reads as follows: "... alterations in cell morphology correlate with restricted trafficking potential" and "... trafficking of macrophage precursors to the embryo correlated with defined stages of their development". In the discussion, the revised section now reads as follows: "Thus, alterations in cell morphology accompanying macrophage maturation correlate with their distribution through the bloodstream. However, the transition from round cells to dendritic macrophages could be happening independently of the trafficking process within tissues."

Comment 8 Please provide a small sentence to expand on the in vivo imaging strategy (line 96). Additionally, adjust the methods to clarify if the embryo is removed from the uterus or attached to the uterus of the live mother; the figure implies the latter.

Between embryonic days E8.5 and E12.5 we used both techniques for our in vivo microscopy approach. We obtained the same results in both experimental settings as long as the microscopy time was limited to 15 minutes. However, movies of embryos isolated from the mother with an intact yolk sac and placenta provided a better imaging quality for recording movies.

The revised method section reads as follows: "Access to the abdominal cavity of pregnant mice was provided by a cesarean section. Depending on the developmental stage, intravital imaging was predominantly carried out in separation from the mother animal (E8.5 to E12.5) for reasons of image quality or in continuous connection to the maternal circulation (E12.5 to E16.5)."

We also provide additional experimental data on the trafficking kinetics of E10.5 embryos connected to the maternal circulation (Supplementary Fig. 1g, h). The data are similar to the results obtained by imaging embryos independently of a connection to the mother animal (Fig. 1e, i). This information was added to the method section: "Quantity and velocity of trafficking Cx3cr1+ cells in separated E10.5 embryos was similar to embryos being in continuous connection to the mother (Supplementary Fig. 1g, h)."

Figure 1: Trafficking of Cx3cr1+ yolk sac pre-macrophages via the bloodstream. (e) Quantification of Cx3cr1+ cells per microscopic field of 400 μm x 400 μm ; (***) $p < 0.001$ (t-test). (i) Quantification of intravascular Cx3cr1+ cells in the YS in an average-sized vessel.

Supplementary Figure 1: The $Cx3cr1^{gfp/+}$ reporter mouse model effectively labels pre-macrophages in the yolk sac. (g) Quantification of intravascular Cx3cr1+ cells in an average sized vessel (in connection to maternal circulation). (h) Velocity of flowing Cx3cr1+ cells in the yolk sac vasculature (in connection to maternal circulation).

Comment 9 In line 138 “Consistent with the trafficking of YS precursors through vascular routes, there was a short time delay of few hours between detecting pre-macrophages in the yolk sac and their subsequent appearance of macrophages in embryonic tissues.” Would there not be a short time delay by tissue-migration method also?

We thank the reviewer for this comment. We absolutely agree and omitted this sentence.

Please provide evidence for direct cell-to-cell contacts (line112).

The reviewer refers to figure 1d. We rephrased the paragraph as follows: “*In parallel to the increase in cell numbers, Cx3cr1+ pre-macrophages underwent morphological changes from spherical shape to mature cells with multiple dendrites (Fig. 1d) potentially engaging in direct cell-to-cell contacts, as it is typically found in adult tissues*²⁴.”

Comment 10 The authors stated in Line 168 “, the primary target for YS-derived macrophages was the developing brain”, however, the manuscript did not contain any data about the kinetics and numbers of macrophages in different organs, so the statement could be clarified.

To confirm our visual impression (Fig. 2a) that Cx3cr1 GFP+ cells were primarily targeted to the brain, we performed additional experiments. We dissected yolk sac, brain and the distal embryonic region (trunk) at different time points of development and carried out flow cytometry analysis. We confirmed the early infiltration of the brain region which equalizes with other embryonic regions in the course of development, i.e. E12.5 (Fig. 2b, c, d). The results are in line with a previous study by Ginhoux et al.¹⁵.

Figure 2: Pre-macrophages infiltrate embryonic tissues. (a) Isolated E10.5 *Cx3cr1^{gfp/+}* embryo. (b, c) Visualization (b) and quantification (c) of *Cx3cr1*⁺ macrophages in different embryonic regions at indicated time points; (*) $p < 0.05$ (ANOVA). (d) Quantification by FACS of *Cx3cr1* GFP⁺ cells in the embryonic yolk sac, brain and trunk at indicated time points; (**) $p < 0.01$ (Mann-Whitney test). (e) *Cx3cr1* GFP⁺ cells (green) in the embryonic brain at E16.5 additionally stained for the endothelial marker CD31 (red). (f) Image series extracted from live E10.5 video sequences (Supplementary Movie 3) showing a cell attaching to the endothelium in the embryonic head region; flow top to bottom. Scale bars are 1mm (a), 100 μ m (b, e, f).

In addition, in Line 170 – “macrophages in the embryonic brain could be frequently visualized in close proximity to vascular structures” but the referenced pictures did not provide evidences for the statement. The resolution is not high enough, the pictures show random distribution of macrophages.

We thank the reviewer for this comment and re-analyzed our data. *Cx3cr1* GFP⁺ macrophages could be visualized in proximity to vascular structures (Fig. 2e). However, we could not identify a clear pattern of macrophage distribution, and random distribution is indeed most likely. We therefore changed the respective sentence in the results section as follows: “*Although some Cx3cr1+ cells were found in close proximity to the brain vasculature, they distributed throughout the head region forming a dense cellular network (Fig. 2b, e and Supplementary Movie 3).*”

Comment 10 Line 174 “Taken together, these data provided direct evidence that YS pre-macrophages trafficked to the embryo proper through the bloodstream and entered respective target tissues via vascular routes.” This is overstated, simultaneous establishment of vessels and pre-macrophage is not direct evidence of the mechanism of trafficking

We thank the reviewer for this comment. We deleted the sentence, which was indeed unsuitable in this context.

Comment 11 We suggest that the authors rephrase “abortion material” (line 182) to human fetal material and would prefer that the specific proportion of the fetus be stated. The source of human fetal material can be stated in the methods.

We rephrased the sentence and wrote human fetal material. The proportion of the fetus (yolk sac or placenta) was labeled in every figure, and the source was added in the methods section.

Comment 12 The authors do not explain the experimental result with CD68 Figure2 in the main text. We request that the authors state that human premacrophages were detected with CD68 to parallel the emphasis on how mouse pre-macrophages were detected and distinguish methods between the two species.

The human fetal material was stained with CD68 as a common macrophage marker in the human system. We performed additional histological analysis and added *Cx3cr1* and F4/80 as macrophage markers, in analogy to our analysis in mice. We conclude from these experiments

that “... macrophages of the human YS expressed CX3CR1 and F4/80 as in mice (Supplementary Fig. 2a, b). Thus, we identified similarities between mouse and human YS macrophages, opening the possibility of analogies in their development.”

Supplementary Figure 2: Human yolk sac macrophages. (a, b) Confocal fluorescence images of human yolk sac and placenta stained for CD68 (green), Cx3cr1 (red, upper left panel in (a)), F4/80 (upper right panel in (a)) and DAPI (blue). Samples were obtained from abortion material, age of gestation: 9 weeks +1 day. Scale bars are 50 μ m.

Comment 13 The experiment referred to in line 226 did not shed any light into the molecular mechanism of trafficking because it was not rationalized on this basis. Again, the authors could clarify their thinking on what molecular mechanisms would be controlled by HSC origins and could helpfully suggest alternative hypotheses.

Since the absence of functional Cx3cr1 was reported to affect macrophage density in early embryonic tissues³⁴, we considered Cx3cr1 as a potential candidate to modulate intravascular trafficking of pre-macrophages. Although we could reproduce the published findings of different cell densities in embryonic tissues, there was no alteration of trafficking dynamics. In the results section, the rationale for this experiment is explained as follows: “... macrophage colonization of the mouse embryo occurs in a Cx3cr1-dependent manner³⁴. We therefore addressed whether pre-macrophage trafficking was modulated by Cx3cr1.”

Comment 14 Consider revision of tense Line 234 “travelled”.

The tense was changed according to the reviewer’s suggestion.

Comment 15 Please state line 237 “Thus, alterations in cell morphology restrict the trafficking potential of YS macrophages.” More accurately, we suggest “...correlate with restricted trafficking potential”

We thank the reviewer for his comment and changed the sentence as suggested.

Comment 16 Line 238 “maturated”, Consider revision to “matured”

The tense was modified as suggested by the reviewer.

Comment 17 The authors state that, “Cx3cr1+ pre-macrophages represented a large fraction of the Csf1r+ EMPs (Figure 6d)” Line 279. This is a qualitative statement and quantitative evaluation is needed to support this finding.

We quantified Cx3cr1+ cells within the population of Csf1r+ pulse-labeled cells on E10.5 and found that approximately 90% co-expressed both markers. This information was added in Suppl. Fig. 4c.

C

FACS quantification of Cx3cr1+ cells among CD45+ Csf1r+ pulse-labeled cells on E10.5 in the yolk sac.

Comment 18 The authors state that “In Csf1rMer-iCre-Mer:Rosa26LSL-YFP embryos in vivo, pulse-labeled EMPs displayed a clonal expansion pattern in tissues” (Line 290 and 298). Based on the data presented, where only one color was used, one cannot draw a conclusion about the clonality. The “rainbow” mice should be used for such

statements.

We thank the reviewer for his comment. It is correct that we cannot prove clonal expansion with our model. However, pictures in the yolk sac strongly suggest a small number of seeding cells that proliferate and lead to the formation of cell clusters. This expansion pattern is also evident in parts of the embryo (Fig. 6e, f). We analyzed our imaging data to determine the size of respective macrophage clusters, and now provide quantitative data on Csfl1⁺ cell clusters in the YS and embryonic tissues. However, we agree with the reviewer regarding the term “clonality” and rephrased the respective sentence omitting this word. The revised text reads as follows: “*In the YS, cell clusters were relatively large (about 20 cells) while in the embryo cell conglomerates mostly consisted of few YFP+ cells (2-7 cells) at E12.5 (Fig. 6e, f)*”.

We also agree with the reviewer that the “rainbow” mouse should be used to proof clonality in this model. We therefore added the following limitation in the revised text: “*Future fate mapping experiments harnessing multicolor reporter mice are necessary to proof clonality of these cell clusters*”.

Comment 19 The authors state that, “**In the YS, clones were relatively large (>20 cells) while in the embryo cell conglomerates mostly consisted of few Csfl1⁺ cells (2-7 cells).**” (Lane – 292-294) **This is an important observation; however, no data is presented to support it.**

Data was quantified and added as a new figure (Fig. 6f).

Comment 20 **The observation of Kupffer cells as derived from Csfr1 pulsed at E8.5 is important, as shown in Figure 6f and g. Did authors observed any other blood cells or other tissues with CSFr1 derived cells in these mice?**

We reported earlier on the fate mapping of YS progenitors by pulse labeling *Csfl1^{MerCreMer}.Rosa26^{eYFP}* embryos at E8.5. This model induces YFP expression in YS-derived Csfr1⁺ cells and their progeny upon application of tamoxifen. In adult mice that have been injected at embryonic day E8.5, macrophages in all organs express YFP, albeit in different quantities, whereas other hematopoietic cells are not labeled^{9, 19}.

Comment 21 **The authors state that, “Thus, alterations in cell morphology accompanying macrophage maturation are less compatible with their distribution through the bloodstream.” (Line 316) While the postulated causation is plausible, it is by no means proved by the available evidence and would thus best be described as a correlation.**

We agree with the reviewer. As discussed in comment 7, we changed the text as follows: “*Thus, alterations in cell morphology accompanying macrophage maturation correlate with their distribution through the bloodstream. However, the transition from round cells to dendritic macrophages could be happening independently of the trafficking process within tissues.*”

Comment 22 Please Quantify and provide statistical evaluation (Line 333) “...and we detected hardly any intravascular Cx3cr1+ macrophages after E12.5.”

Quantitative data is shown in Figure 1i.

Comment 23 On line 347 the authors state, “Furthermore, a significant proportion of immune cells emerge from the YS as more differentiated Cx3cr1+ pre-macrophages.” – This can’t be stated without quantitative analysis, which is missing.

We thank the reviewer for this comment. We rephrased the whole section and omitted this sentence.

Comment 24 The authors should rule out that vessel size affects macrophage trafficking by performing quantification on small, average and large vessels in order to see if vessel size will alter the timeframe of trafficking.

We quantified the size of the imaged vessels and included the new data in supplementary figure 1f.

Average vessel size increases with development as can be expected. From the size of the vessels imaged, it seems unlikely that this parameter affected macrophage trafficking, specifically it would not explain the drop in the number of trafficking macrophages from E12.5. However, we would refrain from making such statement in the present manuscript since data on shear stress and other factors are missing and cannot be quantified easily.

f

Supplementary Figure 1: The *Cx3cr1^{gfp/+}* reporter mouse model effectively labels pre-macrophages in the yolk sac. (f) Quantification of YS vessel size used for trafficking quantifications at indicated time points.

Comment 25 Please refer the reader to the specific movies used to generate the data in the main figure legend

We added a reference to each movie in the respective figure legend.

Comment 26 Provide statistics throughout and describe the statistical test used

Statistics were added in the figures. Information on the respective statistical test was included in the figure legends.

Comment 27 Adjust black bars in bar graphs as they obscure data points

The color scheme has been adjusted.

Comment 28 Arrowheads are far too small

We have increased the size of arrowheads.

Comment 29 It would be preferable to use Definitive, rather than descriptive, figure legend titles. E.g. Figure 2 and 4 legend titles should be revised to follow the style of the others.

We changed the figure legend titles.

Comment 30 Verify that the dotted white lines added to images are indeed vessels by showing co-localization of specific marker staining and state what the dotted white lines represent

The dotted line represents the endothelium. To verify vascular borders we routinely performed CD31 staining to visualize the endothelium.

Figure 1: Trafficking of Cx3cr1⁺ yolk sac pre-macrophages via the bloodstream. (f) *In vivo* YS staining for the endothelial marker CD31 (red) in *Cx3cr1^{gfp/+}* (green) embryos at E10.5. Scale bars is 100 μ m.

Comment 31 Figure 1.**a. Figure 1 c: what is the relevance of this panel**

The authors believe that many readers are not familiar with yolk sac anatomy. Therefore, we would like to illustrate its general appearance, including blood vessels, as well as its anatomical relation to the embryo proper. If desired, this panel can be removed.

b. Figure 1d and f: these panels show the same result

We thank the reviewer for this comment. We have revised the figure accordingly.

c. Figure 1 e: What is the error bar? What is the black bar representing? Illustrate the black bar in a more transparent way as it is obscuring the data points. Provide statistical evaluation

Statistics were added and color scheme was adjusted.

d. Figure 1g: Arrow heads are far too small throughout

Size of arrowheads were adjusted.

e. Figure 1h. What is 30+1s label? What embryonic stage is this at?

The label 30+1s illustrates that this intravital microscopy image was obtained just one second after the previous one. We changed it to 31s for a better understanding. The embryonic stage is E10.5.

f. Figure 1h. shows cells outside of the vessel wall and not entering

The process of entering happens from one second to the next, as initially illustrated by the 30+1s label. As soon as the cell enters the blood stream it leaves the microscopic field. To further clarify this process of entering, we provided a representative movie (Supplementary Movies 1a, b).

g. Figure 1j. What embryonic stage is that at?

This figure refers to embryonic day E10.5. We added this information to the respective figure.

h. Figure 1k. what is the biological significance of flowing vs rolling

Cells trafficking with the flowing blood in an undisturbed manner were indicated as “flowing” cells. Mean velocity of flowing cells was $>200 \mu\text{m}/\text{second}$ in our experiments. However, we agree that this term does not clearly indicate its biological significance. We

therefore changed the term “flowing” to “non-adhering” cells, which is more commonly used in the literature. In vascular biology, cell rolling is defined as a process of transient interactions with the vascular wall leading to a reduction in trafficking velocity^{35, 36}. Consequently, velocity of rolling cells was lower than that of non-adhering cells (mean velocity was 25 $\mu\text{m}/\text{second}$). The biological significance of these transient interactions with the wall of yolk sac vessels is, however, unknown.

i. Legend. “Line 699 (c) E16.5 Cx3cr1gfp/+ embryos with surrounding YS and its dense vascular network connected to the placenta. (d, f).” Consider revision as the YS vasculature is not connected to the placenta the YS vasculature is connected to the embryo and the embryo is connected to the placenta.

In figure 1c, the sentence was changed as follows: “... *embryo with surrounding YS and its dense vascular network*”.

j. Legend. What is an average vessel size?

We quantified the size of the imaged vessels and included the new data in supplementary figure 1f. As can be seen, average size of the imaged vessels is approximately 60 μm at E9.5 and increases with development as can be expected.

Comment 32 Figure2.

a. Figure 2 c. what statistical test was performed?

After positive testing for normal distribution (D’Agostino & Pearson) we performed a one-way ANOVA with Tukey’s multiple comparison test. The information on the statistical tests applied as well as their results were added throughout the manuscript.

b. State what CD31 stains for

It is an endothelial marker. This information was added in the method section and the figure legends.

c. State what CD68 stains for

CD68 is a macrophage marker, which is commonly used in the human system. We added this information in the results section.

d. Gestational age is +2 weeks not +1 week.

In medicine the label “week x + y” is commonly used: “y” represents the time in days beyond “week x”. The gestational age is now indicated more clearly as “9 weeks + 1 day” (legend to supplementary figure 2).

Comment 33 Figure3.

a. Figure 3C. At what stage of development is this?

The graph refers to embryonic day E12.5. We added this information to the respective figure.

b. Figure 3d. no distinction between color of bar graphs

The bar graphs have been adjusted.

c. Figure 3 d,e,f,g Authors should include a statement of statistical significance and state the statistical test used for all data

The information on the statistical tests applied as well as their results were added throughout the manuscript.

d. Line 738 “Scale bar is 100µm (d). “d” should be “c”

We thank the reviewer for his remark. The label was corrected.

Comment 34 Figure 5d. what stage of development?

This figure refers to embryonic day E9.5. We added this information to the respective figure.

Comment 35 Figure 6.

a. Figure 6f and g. Far too small to determine cellular resolution of staining.

Higher resolution pictures have been provided.

b. Figure 6f and g. What is the biological significance of the images?

Figures 6f and 6g illustrate the persistence of E8.5 pulse-labeled yolk sac macrophages and their progeny in the adult mouse (1 year old). It also illustrates that clusters of macrophages can persist over time.

As stated above, we agree with the reviewer that the “rainbow” mouse should be used to proof clonality in this model. We therefore added the following limitation in the revised text: *“Future fate mapping experiments harnessing multicolor reporter mice are necessary to proof clonality of these cell clusters”*.

Comment 36 Movie S6a and 6b. Query that the scale bar is the same in each movie. The resolution of 6b is too poor to determine the result.

Comment 37 Movie S7a and 7b. Query that the scale bar is the same in each movie. The resolution of 7b is too poor to determine the result. If the scale bars are correct that the comparison is between different size vessels at each developmental age of analysis.

- 1) We checked the scale bars and confirmed their correctness.
- 2) The yolk sac is quite difficult to image due to its autofluorescence. In movie 2b (E12.5 Cx3cr1GFP+ embryo) intravascular GFP+ cells are extremely rare, whereas in movies 7b (E12.5 *Csf1r*^{Cre}:*Rosa26*^{eYFP} embryo) and 8b (E12.5 *Csf1r*^{MerCreMer}:*Rosa26*^{eYFP} embryo + Tamoxifen pulse) trafficking YFP+ cells could still be detected. By analyzing movies in detail, including frame-by-frame analyses, we were able to clearly identify YFP+ cells in the bloodstream.

Reviewer #2 (Remarks to the Author):

It is now will recognized that tissue-resident macrophage populations in many adult organs are ultimately derived from hematopoiesis in the yolk sac prior to the emergence of hematopoietic stem cells (in no small part by the pioneering studies of the authors). In this paper, the authors examine the migration of macrophage precursors and hematopoietic progenitors, so-called EMP, from the yolk sac into the embryo proper.

The experiments are well described and the data are logically laid out. The authors provide convincing data, through images and multiple movies, that Cx3cr1+ macrophage cells that arise in the yolk sac enter the bloodstream and move into the embryo to seed various organs, starting with the head region. Use of genetically-labeled cells make these elegant tracking studies possible. Evidence that macrophage cells use the vasculature to migrate into the embryo is the main finding of this paper. While, as authors point out, the route macrophages take to colonize embryonic tissues has been controversial, recent studies of Ncx1-/- mouse embryos support the role of the vasculature, particularly Ginhoux, et al. *Science* (2010), as cited by the authors, as well as Hoeffel, et al. *JEM* (2012), that shows a complete absence of macrophages in the skin of Ncx1-/- embryos. Ncx1-/- embryos also have normal/increased numbers of EMP in the yolk sac but lack EMP in the embryo (Lux, et al. *Blood* (2008), Frame, et al. *Stem Cells* (2016)). The normal/increased numbers of macrophage cells (this paper) and EMP in the yolk sac and the essentially complete lack of these cell types in the embryo of mouse embryos lacking a functional circulation provide strong presumptive evidence that macrophage cells and EMP seed the embryo through the circulation. The positive results presented here corroborate this conclusion with positive data.

We thank the reviewer for this positive feedback on our manuscript. Indeed, the combination of microscopy with genetic labeling allowed for conducting these tracking studies in mouse embryos *in vivo* and provided further insight into the development of the innate immune system.

We also thank the reviewer for the comment on the trafficking routes of macrophages. We have made in-depth changes in several parts of the manuscript during the revision process. We also included above references to more thoroughly cover the previous work on Ncx1-/- embryos. Further, the fact that our imaging model is suited for studying fast intravascular processes was added as a limitation regarding the analysis of trans-tissue migration. The revised paragraph in the discussion section now reads as follows: *“By applying intravital microscopy we provided direct evidence that YS macrophage progenitors colonize the embryo proper via the vasculature. There has been an ongoing debate on how YS macrophages enter the embryo proper, that is, by trans-tissue migration or trans-vascular trafficking via the bloodstream. The early appearance of myeloid colony-forming units in embryonic blood before the onset of fetal definitive hematopoiesis indicates trans-vascular trafficking²¹. Further, infiltration of embryonic tissues by EMPs and maturing macrophages is impaired in mice lacking a heartbeat supporting the role of a functional circulation for macrophage trafficking^{11, 15, 37, 38}. However, this model is limited by its dramatic impairment of overall mouse physiology resulting in early embryonic lethality³⁹. In contrast to data supporting trans-vascular migration, it has been suggested for various species that macrophages can colonize embryonic tissues independently of blood vessels via extravascular routes^{40, 41, 42, 43}. In line with this, functional circulation is not required for EMP emergence in the YS¹¹. Our data indicate that intravascular trafficking is the predominant way of macrophage progenitors infiltrating the embryo proper. However, the microscopy tools applied in this*

paper are suited for imaging fast intravital processes and, therefore, we cannot exclude nor quantify trans-tissue migration.”

In the introduction, the authors state that adult tissue-resident macrophage cells “develop independently of definitive hematopoiesis from EMPs in the yolk sac”. While this is true, it is also evident that CFU-Mac arise in the yolk sac before EMP and that differentiating macrophage cells are present in the yolk sac as EMP emerge, as is evident at E9.5 (Frame, et al. Stem Cells 2016) and at E10.25 (this paper, Figure 6d, where several Csf1R+, Cx3cr1+, F4/80+ cells are evident). In addition, lineage tracing studies in mice indicate that microglia arise from different yolk sac progenitors than the rest of the tissue-resident populations in the embryo. These findings indicate that the underlying model system that the authors use to interpret their data (that only EMP in the yolk sac give rise to all pre-macs) does not do justice to the complexity of hematopoiesis in the yolk sac.

We completely agree with the reviewer that the complexity of YS hematopoiesis was not presented clearly enough and that the Csf1r-dependent model did not allow to track YS EMPs. We therefore performed new fate mapping and flow cytometry experiments, and revised various sections of the main text.

1) We revised the introduction section and included a description of the different waves of hematopoiesis. Specifically, we introduced the first (primitive) wave leading to the formation of macrophages that primarily traffic to the brain.

The revised paragraph in the introduction reads as follows: “... *first macrophage precursors develop in the YS, the first site of hematopoiesis in mammals*²⁴. *Fate mapping analyses and lineage tracing experiments recently indicated that tissue macrophages in many organs are of early embryonic origin*^{13, 15, 19, 44, 45}. *Macrophages develop in the YS in two successive waves: First, primitive macrophage progenitors appear around E7.25 in the YS and predominantly infiltrate the brain after the onset of circulation*^{15, 20, 21, 22}. *Second, multipotent erythromyeloid progenitors (EMPs) develop from E8.25 onwards and represent a major source of tissue resident macrophages*^{9, 23, 24}. *Significantly later, starting from E10.5, hematopoietic stem cells (HSCs) arise in the aorto-gonado-mesonephros (AGM) region and migrate to the liver to initiate fetal definitive hematopoiesis which later shifts to the BM*^{25, 26, 27}. *In adult life, ...*”

2) In addition, we performed flow cytometry of mouse embryos between E8.5 and E10.5 to determine the phenotype of YS hematopoietic progenitors. We identified Kit- CD16/32+ cells at E8.5, while only a very small number of Kit+ cells was present at this time (Supp. fig. 5). Kit- CD16/32+ Cx3cr1- cells were previously described as progenitors with unipotent primitive macrophage potential. Their numbers decreased progressively towards E12.5. Kit+ CD16/32+ Cx3cr1+ appeared around E9.5 and strongly increased towards E10.5.

We summarized the results as follows: “*Flow cytometry analyses of the YS revealed a CD16/32+ Kit- population of primitive macrophages progenitors before the first appearance of EMPs suggesting their independent origin in line with previous reports (Supplementary Fig. 5a)*^{8, 20}. *Only a few hours later Kit+ EMPs develop in the YS and seed the fetal liver (Supplementary Fig. 5a-d) before they differentiate into Kit- Cx3cr1+ pre-macrophages as described previously (Supplementary Fig. 5b, e)*^{8, 20}. *To more specifically determine the trafficking of YS-derived EMPs we carried out intravital microscopy in mice, in which Cre-dependent recombination was under control of the Kit promoter ...*”.

Supplementary Figure 5: Primitive YS macrophages appear before the first detection of Kit⁺ cells. (a) FACS plots of CD16/32⁺ Kit⁻ (population I) and Kit⁺ (population II) cells in the embryonic yolk sac at indicated time points. (b-e) FACS plots (b) and corresponding quantifications (c-e) of Kit⁺ CD16/32⁺ progenitors in the yolk sac (c) and fetal liver (d) as well as Kit⁻ CD16/32⁺ Cx3cr1 GFP⁺ cells in the yolk sac (e) at indicated time points. FACS plots show an individual representative experiment. At least 3 embryos of minimum 2 independent litters were analyzed per time point.

3) To more specifically address the trafficking of EMPs, we harnessed their expression of Kit⁸ and carried out pulse labeling experiments in *Kit^{MerCreMer}·Rosa26^{eYFP}* embryos. While the Csf1r-dependent model is expected to label both EMPs and maturing macrophages of primitive hematopoiesis, pulse labeling in *Kit^{MerCreMer}* embryos allowed us to more specifically address the trafficking of YS EMPs and their progeny. The data was added in new figure 7. The new results section reads as follows: “To more specifically determine the trafficking of YS-derived EMPs we carried out intravital microscopy in mice, in which Cre-dependent recombination was under control of the *Kit* promoter. *Kit* has previously been

identified as a marker of EMPs^{20, 32}, and distinguishes them from maturing YS macrophages and primitive progenitors^{8, 11}. In order to label *Kit*⁺ progenitors, we crossed *Kit*^{MerCreMer} mice with *Rosa26*^{mT/mG} reporter mice and applied a single dose of OH-TAM to induce GFP expression (Fig. 7a). In *Kit*^{MerCreMer}:*Rosa26*^{mT/mG} embryos, OH-TAM application from E7.5 to E8.5 resulted in increased labeling of EMPs (Fig. 7b-d). Intravital microscopy at E10.5 showed that numerous EMPs, or their progeny, migrated to the embryo proper via the YS vasculature (Fig. 7e, f and Supplementary Movie 9). *Kit*⁺ cells thereby infiltrated all embryonic regions including the brain (Fig. 7b, c and g).”

Figure 7: Trafficking of *Kit*⁺ EMPs. (a) Schematic graph for the *Kit*^{MerCreMer}:*Rosa26*^{mT/mG} mouse model with TAM pulse labeling at E7.5 or E8.5 (red arrows) and subsequent analyses at E10.5 (black arrow). (b, c, d) Fluorescent pictures of *Kit*⁺ cells in the YS and embryo

proper on E10.5 after pulse-labeling at indicated time points (b, c) with corresponding quantifications (d); (***) $p < 0.001$ (Mann-Whitney test). (e) Quantification of labeled Kit⁺ cells/ minute in an average-sized YS vessel at E10.5. (f) Velocity of intravascular non-adhering Kit⁺ cells in the YS at E10.5. (g) Flow cytometry plots of CD45⁺ gated Kit⁺ versus pulse-labeled Kit^{MerCreMer}:Rosa26^{mT/mG}⁺ cells in YS, head and trunk region at E10.5. Plots show an individual representative experiment. At least 3 embryos of minimum 2 independent litters were analyzed per time point. Scale bars are 100 μ m (b), 1 mm (c).

We believe that the combination of model systems applied in this manuscript, allows us to more adequately interpret the data of macrophage trafficking in light of the complexity of YS hematopoiesis.

The authors make use of an inducible Csf1R to mark EMP and conclude that EMP also use the circulation to enter the embryo- a conclusion consistent with studies in normal, as well as Ncx1^{-/-} embryos (as mentioned above). However, induced Csf1r-labeling at E8.5 would simultaneously mark not only some/most EMP but also the differentiating macrophage cells from the CFU-Mac that arise prior to EMP. I suspect it is these macrophage cells that are evident as Csf1R⁺, Cx3cr1⁺, F4/80⁺ cells in Figure 6d, and possibly to most of the CX3cr1⁺ pre-macrophage cells in the yolk sac, and which seed the head (Figure 2a) to generate the permanent microglia.

In line with this, the number of trafficking Csf1r⁺ cells per minute is nearly the same as the Cx3cr1⁺ pre-macrophages at E9.5 (Figures 1i and 5c). The difference at E12.5 would reflect Csf1R-labeling of EMP vs the differentiating "primitive" macrophage cells, which have migrated out of the yolk sac. These data also suggest that EMP, which peak at E10-10.5 in the yolk sac and are still trafficking at E12.5 (Fig. 5c), give rise to very few and no trafficking pre-macrophage cells at E11.5 and E12.5, respectively (Fig. 1i).

We thank the reviewer for this comment. As presented above, we carried out additional fate mapping analyses in *Kit^{MerCreMer}:Rosa26^{mT/mG}* embryos to address the trafficking of YS EMPs and their progeny. The new data was included in figure 7. Interestingly, the number of trafficking cells labeled in E8.5 tamoxifen-pulsed *Kit^{MerCreMer}* embryos was lower than in E8.5 tamoxifen-pulsed *Csf1r^{MerCreMer}* mice at E10.5 (when trafficking reached its maximum). The difference might be explained by labeling of both EMPs and differentiating "primitive" macrophages in the Csf1r-dependent model. In the revised discussion section, we therefore included the following paragraph: *"The observation that more fluorescent cells were trafficking in E10.5 Csf1r^{MerCreMer} embryos than in Kit^{MerCreMer} mice might be explained by the labeling of both maturing macrophages derived from primitive hematopoiesis and EMPs in the Csf1r-dependent model, whereas pulse labeling in Kit^{MerCreMer} embryos is more specific for fate mapping of EMPs⁸."*

Further, we agree with the reviewer that the difference in the number of trafficking cells at E12.5 between pulse-labeled *Csf1r^{MerCreMer}* and *Cx3cr1^{gfp/+}* mice likely reflects that of Csf1r⁺ EMPs and their progeny. We added this comment in the discussion section, which reads as follows: *"The difference in the number of trafficking cells at E12.5 between pulse-labeled Csf1r^{MerCreMer} (Fig. 5, Fig. 6f) and Cx3cr1^{gfp/+} (Fig. 1i) mice could reflect Csf1r-labeling of EMPs as compared to matured "primitive" macrophages, the latter having completed their emigration from the YS at this time."* Further, we added a flow cytometry analysis of E12.5 fetal liver demonstrating the labeling of macrophages, erythrocytes and neutrophils, but not lymphoid cells, in pulse-labeled *Csf1r^{MerCreMer}* embryos (Suppl. Figure 4).

Supplementary Figure 4: *Csf1r* pulse labeling model tracks EMPs. (a) Schematic graph for the *Csf1r^{MerCreMer}:Rosa26^{eYFP}* mouse model with TAM pulse labeling. (b) Analysis of labeling efficiency of erythroid and myeloid lineages in E12.5 fetal liver. (c-d) Flow cytometry analysis of *Csf1r^{MerCreMer}:Rosa26^{eYFP}* pulse-labeled cells. (c) Quantification of Cx3cr1-expressing cell among YFP+ cells in E10.5 YS. (d) Surface expression of indicated lymphoid markers in at E12.5 in the YS and fetal liver. Flow cytometry plots show an individual representative experiment. At least 3 embryos of minimum 2 independent litters were analyzed per time point.

Studies conducted in Myb^{-/-} embryos, which lack hematopoietic stem cell function, confirm that the identified macrophage kinetics are independent of Myb gene expression. These latter findings are not unexpected given the pioneering studies published previously by the authors (Schulz, et al. 2012) that Myb^{-/-} embryos have normal numbers of macrophage cells in the embryo.

We agree with the reviewer that the findings in Myb mice were not completely unexpected. However, in the previous paper we analyzed macrophage numbers in E14.5 and E16.5 Myb^{-/-} embryos¹⁹. As macrophage trafficking was mostly completed at E12.5, macrophage numbers might have equilibrated in tissues (i.e. between E12.5 and E14.5), even if there was an impact on kinetics. We therefore believe that it was important to address macrophage trafficking at E12.5, when the number of migrating cells decreased strongly.

Minor points:

1. Is it better to show the data as bar graphs in Figure 1i?

We thank the reviewer for this comment. We have changed the figure to include bar graphs.

2. Page 6, last 2 sentences of the middle paragraph: Is there a difference between “macrophage precursors” and “macrophage progenitors”?

There is no difference. We now use the term “macrophage progenitors” throughout the manuscript.

References

1. Moore, M.A. & Owen, J.J. Chromosome marker studies in the irradiated chick embryo. *Nature* **215**, 1081-1082 (1967).
2. Moore, M.A.S. & Owen, J.J.T. STEM-CELL MIGRATION IN DEVELOPING MYELOID AND LYMPHOID SYSTEMS. *The Lancet* **290**, 658-659.
3. Lassila, O., Eskola, J., Toivanen, P., Martin, C. & Dieterlen-Lievre, F. The origin of lymphoid stem cells studied in chick yolk sac-embryo chimaeras. *Nature* **272**, 353-354 (1978).
4. Lassila, O. *et al.* Is the yolk sac the primary origin of lymphoid stem cells? *Transplantation proceedings* **11**, 1085-1088 (1979).
5. Dieterlen-Lievre, F. On the origin of haemopoietic stem cells in the avian embryo: an experimental approach. *Journal of embryology and experimental morphology* **33**, 607-619 (1975).
6. Huang, H., Zettergren, L.D. & Auerbach, R. In vitro differentiation of B cells and myeloid cells from the early mouse embryo and its extraembryonic yolk sac. *Experimental hematology* **22**, 19-25 (1994).
7. Liu, C.P. & Auerbach, R. In vitro development of murine T cells from prethymic and pre-liver embryonic yolk sac hematopoietic stem cells. *Development* **113**, 1315-1323 (1991).
8. McGrath, K.E. *et al.* Distinct Sources of Hematopoietic Progenitors Emerge before HSCs and Provide Functional Blood Cells in the Mammalian Embryo. *Cell Rep* **11**, 1892-1904 (2015).
9. Gomez Perdiguero, E. *et al.* Tissue-resident macrophages originate from yolk-sac-derived erythro-myeloid progenitors. *Nature* **518**, 547-551 (2015).
10. Boiers, C. *et al.* Lymphomyeloid contribution of an immune-restricted progenitor emerging prior to definitive hematopoietic stem cells. *Cell stem cell* **13**, 535-548 (2013).
11. Frame, J.M., Fegan, K.H., Conway, S.J., McGrath, K.E. & Palis, J. Definitive Hematopoiesis in the Yolk Sac Emerges from Wnt-Responsive Hemogenic Endothelium Independently of Circulation and Arterial Identity. *Stem cells* **34**, 431-444 (2016).
12. Jung, S. *et al.* Analysis of fractalkine receptor CX(3)CR1 function by targeted deletion and green fluorescent protein reporter gene insertion. *Molecular and cellular biology* **20**, 4106-4114 (2000).
13. Yona, S. *et al.* Fate mapping reveals origins and dynamics of monocytes and tissue macrophages under homeostasis. *Immunity* **38**, 79-91 (2013).

14. Ovchinnikov, D.A. *et al.* Expression of Gal4-dependent transgenes in cells of the mononuclear phagocyte system labeled with enhanced cyan fluorescent protein using Csf1r-Gal4VP16/UAS-ECFP double-transgenic mice. *Journal of leukocyte biology* **83**, 430-433 (2008).
15. Ginhoux, F. *et al.* Fate mapping analysis reveals that adult microglia derive from primitive macrophages. *Science* **330**, 841-845 (2010).
16. McGrath, K.E., Koniski, A.D., Malik, J. & Palis, J. Circulation is established in a stepwise pattern in the mammalian embryo. *Blood* **101**, 1669-1676 (2003).
17. Koushik, S.V. *et al.* Targeted inactivation of the sodium-calcium exchanger (Ncx1) results in the lack of a heartbeat and abnormal myofibrillar organization. *FASEB journal : official publication of the Federation of American Societies for Experimental Biology* **15**, 1209-1211 (2001).
18. Ji, R.P. *et al.* Onset of cardiac function during early mouse embryogenesis coincides with entry of primitive erythroblasts into the embryo proper. *Circulation research* **92**, 133-135 (2003).
19. Schulz, C. *et al.* A lineage of myeloid cells independent of Myb and hematopoietic stem cells. *Science* **336**, 86-90 (2012).
20. Bertrand, J.Y. *et al.* Three pathways to mature macrophages in the early mouse yolk sac. *Blood* **106**, 3004-3011 (2005).
21. Palis, J., Robertson, S., Kennedy, M., Wall, C. & Keller, G. Development of erythroid and myeloid progenitors in the yolk sac and embryo proper of the mouse. *Development* **126**, 5073-5084 (1999).
22. Sheng, J., Ruedl, C. & Karjalainen, K. Most Tissue-Resident Macrophages Except Microglia Are Derived from Fetal Hematopoietic Stem Cells. *Immunity* **43**, 382-393 (2015).
23. Gomez Perdiguero, E., Schulz, C. & Geissmann, F. Development and homeostasis of "resident" myeloid cells: The case of the microglia. *Glia* **61**, 112-120 (2013).
24. McGrath, K.E., Frame, J.M. & Palis, J. Early hematopoiesis and macrophage development. *Seminars in immunology* (2016).
25. Bertrand, J.Y. *et al.* Haematopoietic stem cells derive directly from aortic endothelium during development. *Nature* **464**, 108-111 (2010).
26. Boisset, J.-C. *et al.* In vivo imaging of haematopoietic cells emerging from the mouse aortic endothelium. *Nature* **464**, 116-120 (2010).
27. Kissa, K. & Herbomel, P. Blood stem cells emerge from aortic endothelium by a novel type of cell transition. *Nature* **464**, 112-115 (2010).
28. Kanitakis, J., Petruzzo, P. & Dubernard, J.M. Turnover of epidermal Langerhans' cells. *The New England journal of medicine* **351**, 2661-2662 (2004).

29. Hoeffel, G. *et al.* C-Myb(+) erythro-myeloid progenitor-derived fetal monocytes give rise to adult tissue-resident macrophages. *Immunity* **42**, 665-678 (2015).
30. Ginhoux, F. & Guilliams, M. Tissue-Resident Macrophage Ontogeny and Homeostasis. *Immunity* **44**, 439-449 (2016).
31. van Berlo, J.H. *et al.* c-kit⁺ cells minimally contribute cardiomyocytes to the heart. *Nature* **509**, 337-341 (2014).
32. Kierdorf, K. *et al.* Microglia emerge from erythromyeloid precursors via Pu.1- and Irf8-dependent pathways. *Nature neuroscience* **16**, 273-280 (2013).
33. Kieusseian, A., Brunet de la Grange, P., Burlen-Defranoux, O., Godin, I. & Cumano, A. Immature hematopoietic stem cells undergo maturation in the fetal liver. *Development* **139**, 3521-3530 (2012).
34. Mass, E. *et al.* Specification of tissue-resident macrophages during organogenesis. *Science* (2016).
35. McEver, R.P. & Zhu, C. Rolling cell adhesion. *Annu Rev Cell Dev Biol* **26**, 363-396 (2010).
36. Chang, K.C., Tees, D.F. & Hammer, D.A. The state diagram for cell adhesion under flow: leukocyte rolling and firm adhesion. *Proceedings of the National Academy of Sciences of the United States of America* **97**, 11262-11267 (2000).
37. Hoeffel, G. *et al.* Adult Langerhans cells derive predominantly from embryonic fetal liver monocytes with a minor contribution of yolk sac-derived macrophages. *The Journal of experimental medicine* **209**, 1167-1181 (2012).
38. Lux, C.T. *et al.* All primitive and definitive hematopoietic progenitor cells emerging before E10 in the mouse embryo are products of the yolk sac. *Blood* **111**, 3435-3438 (2008).
39. Cho, C.H., Kim, S.S., Jeong, M.J., Lee, C.O. & Shin, H.S. The Na⁺-Ca²⁺ exchanger is essential for embryonic heart development in mice. *Molecules and cells* **10**, 712-722 (2000).
40. Herbomel, P., Thisse, B. & Thisse, C. Zebrafish early macrophages colonize cephalic mesenchyme and developing brain, retina, and epidermis through a M-CSF receptor-dependent invasive process. *Developmental biology* **238**, 274-288 (2001).
41. Fantin, A. *et al.* Tissue macrophages act as cellular chaperones for vascular anastomosis downstream of VEGF-mediated endothelial tip cell induction. *Blood* **116**, 829-840 (2010).
42. Chan, W.Y., Kohsaka, S. & Rezaie, P. The origin and cell lineage of microglia: new concepts. *Brain research reviews* **53**, 344-354 (2007).

43. Cuadros, M.A., Martin, C., Coltey, P., Almendros, A. & Navascues, J. First appearance, distribution, and origin of macrophages in the early development of the avian central nervous system. *The Journal of comparative neurology* **330**, 113-129 (1993).
44. Hashimoto, D. *et al.* Tissue-Resident Macrophages Self-Maintain Locally throughout Adult Life with Minimal Contribution from Circulating Monocytes. *Immunity* **38**, 792-804 (2013).
45. Epelman, S., Lavine, K.J. & Randolph, G.J. Origin and functions of tissue macrophages. *Immunity* **41**, 21-35 (2014).

REVIEWERS' COMMENTS:

Reviewer #1 (Remarks to the Author):

This is a great paper. Here are my comments.

Reviewers comments to Rebuttal_ Stremmel et al.,

The authors address a major biological question in the field, when, where, and how 'resident' macrophage precursors traffic from the yolk sac to fetal tissues during development. This is a highly significant and experimentally thorough body of work the authors have contributed not only in their revised manuscript but also in their comprehensive and highly satisfactory rebuttal. In short, I support publication of these findings in Nature Communications as is, or, if the authors wish, following a few minor edits based on comments below.

Minor Comments are numbered as per the original numbered comments as maintained and answered in the authors rebuttal.

A general issue of minor importance; both *kit* and *myb* are expressed by all hematopoietic stem and progenitor cells in mice, not just HSC, and so the absence of cells or the lineage tracing of cells using these genes as promoter/enhancer drivers of genes such as *gfp* or *cre-er* or *mer* does not test HSC only.

Line 121 of revised manuscript: Consider revising, "To resolve this issue".

The authors explore and characterize trans-vascular trafficking. There is no doubt that the authors show trans-vascular trafficking. However, as they point out in the discussion, they do not know the extent to which their results shown represent all the trafficking. It is their call.

Comment 3/ Line 379 in revised manuscript: "Our data indicate that intravascular trafficking is the predominant way of macrophage progenitors infiltrating the embryo proper. However, the microscopy tools applied in this paper are suited for imaging fast intravital processes and, therefore, we cannot quantify nor exclude trans-tissue migration"

The authors have not compared both modalities of trafficking (trans-vascular and trans-tissue) and so to state that the intravascular trafficking is the predominant is a very strong conclusion that they may wish to modulate. The authors might wish to phrase it something like "We interpret our data to indicate that macrophage progenitors infiltrate the embryo proper predominantly by intravascular trafficking. However, the microscopy tools applied in this paper are suited for imaging fast intravital processes and, therefore, we cannot quantify nor exclude trans-tissue migration"

Line 414: insert with between line and the

Line415. Insert h to change were to where

To return to an issue the authors addressed in the introduction and discussion, the authors state that HSC first arise in the AGM on E10.5, and that the earlier yolk sac hematopoiesis is erythromyeloid [EMP]. That is just one side of the story. Moore and Owen first postulated yolk sac origin of full hematopoiesis and did some cell transfers to provide evidence. Weissman, Papiouannou and Gardner published in the 1970s that E8-9 yolk sac blood island cell suspensions transplanted into the yolk sacs of same age hosts distinguishable by H2 alleles and Thy 1 alleles, gave rise to both cfu-s and T cells in the adult hosts. Ueno et al demonstrated by several lineage tracing techniques that individual yolk sac blood islands do not have consistent lineage-traced endothelial cells and blood cells from the same precursor, bringing into question the identity of the

hemangioblast or hemogenic endothelium as an exclusive source of hematopoietic stem/progenitor cells, or that the blood and vascular common precursor exists before the formation of blood islands. Samalkhov and Nishikawa showed by tamoxifen pulsed E7-8.5 runx1Cre-Er mice that the resulting hematopoiesis found in adult marrow and fetal liver derived from these cells includes all blood cell lineages. Inlay et al and Mikkola et al recently provided evidence that before E10.5 yolk sac blood islands at all time points have many more phenotypically defined HSC that transplant full multilineage hematopoiesis in newborn hosts, and that these early HSC are Lyve1+, whereas the AGM never is Lyve1+. All in all, it is obvious that the complexity of yolk sac and post yolk sac hematopoiesis from non-hematopoietic precursors during embryogenesis and fetal life is not yet clear, and that single cell tracking methods will be required to understand just how diverse the early hematopoietic cell populations are, and how many of them participate just during the embryo/fetus interval vs throughout life. The finding by these authors and another group that the resident macrophages derived from yolk sac origins obviously proliferate *pari passu* with the tissues they inhabit as the tiny fetus grows to a ~25 gm adult confound the idea that all hematogenous cells are recently derived from HSC. This paper shows conclusively that csf1R and cxcr3R + precursors give rise to yolk sac origin macrophage lineage cells that, for about 4 days, emigrate massively out of the yolk sac, are found in the yolk sac associated vessels, and are found in tissues throughout the body soon after, and for as long as the mouse lives, or at least was assayed. Therefore it would seem the authors wouldn't want to bias this excellent paper with citing only one side of the HSC story, which is not really relevant to this paper.

Reviewer #2 (Remarks to the Author):

It is now well recognized that tissue-resident macrophage populations in many adult organs are ultimately derived from hematopoiesis in the yolk sac prior to the emergence of hematopoietic stem cells (in no small part by the pioneering studies of the authors). In this revised paper, the authors have responded systematically to all of the reviewers' comments/critique with new experimental data and appropriate modifications to the text. Most importantly, they have added tracking data using Kit expression to more specifically track EMP and to distinguish them from macrophage precursors that initially arise prior to EMP in the yolk sac and go on to seed the developing brain.

The experiments are well described and the data are logically laid out. The authors provide convincing data, through images and multiple movies, that macrophage cells that arise in the yolk sac enter the bloodstream and move into the embryo to seed various organs, starting with the head region. Use of genetically-labeled cells make these elegant tracking studies possible. Evidence that macrophage precursors and EMP both use the vasculature to migrate into the embryo is the main finding of this paper.

Submission number NCOMMS-17-07886A

Rebuttal letter for the manuscript entitled "Yolk sac macrophage progenitors traffic to the embryo during defined stages of development"

Reviewer #1:

This is a great paper. Here are my comments. The authors address a major biological question in the field, when, where, and how ‘resident’ macrophage precursors traffic from the yolk sac to fetal tissues during development. This is a highly significant and experimentally thorough body of work the authors have contributed not only in their revised manuscript but also in their comprehensive and highly satisfactory rebuttal. In short, I support publication of these findings in Nature Communications as is, or, if the authors wish, following a few minor edits based on comments below.

We thank the reviewer for this very positive feedback. In the following, we addressed all comments raised.

A general issue of minor importance; both kit and myb are expressed by all hematopoietic stem and progenitor cells in mice, not just HSC, and so the absence of cells or the lineage tracing of cells using these genes as promoter/enhancer drivers of genes such as gfp or cre-er or mer does not test HSC only.

We completely agree with the reviewer that absence of KIT or MYB is expected to affect both HSCs and their progeny. To address the reviewer’s comment on the expression of MYB in hematopoietic stem and progenitor cells we rephrased the following sentences in the results section:

1) *“We therefore revisited macrophage development in mice lacking fetal definitive hematopoiesis due to genetic absence of the transcription factor MYB (Fig. 4a, b). From E9.5*

through E16.5 CX₃CR1⁺ cell numbers and densities in YS and embryonic tissues were not altered in the absence of HSCs and their progeny”.

2) “These data also showed that most CX₃CR1 GFP⁺ cells identified in tissues after the onset of fetal hematopoiesis were derived from YS hematopoiesis but not from MYB-dependent hematopoietic progenitors.”

Regarding the expression of KIT in hematopoietic stem and progenitor cells we already applied the term “progenitor cells” in a general manner in this paper, e.g. when testing the lineage potential of KIT⁺ cells. For example, we phrased a sentence in the discussion section as follows: “Cells derived from Kit-expressing progenitors thereby infiltrated all embryonic regions including the brain.” We believe that these expressions adequately take into account the reviewer’s comment.

Line 121 of revised manuscript: Consider revising, “To resolve this issue”.

We revised the sentence as follows: “To address these two hypotheses, we directly imaged the trafficking routes of YS macrophage progenitors during embryogenesis using intravital microscopy.”

The authors explore and characterize trans-vascular trafficking. There is no doubt that the authors show trans-vascular trafficking. However, as they point out in the discussion, they do not know the extent to which their results shown represent all the trafficking. It is their call. Comment 3/ Line 379 in revised manuscript: “Our data indicate that intravascular trafficking is the predominant way of macrophage progenitors infiltrating the embryo proper. However, the microscopy tools applied in this paper are suited for imaging fast intravital processes and, therefore, we cannot quantify nor exclude trans-tissue migration.” The authors have not compared both modalities of trafficking (trans-vascular and trans-tissue) and so to state that the intravascular trafficking is the predominant is a very strong conclusion that they may wish to modulate. The authors might wish to phrase it something like “We interpret our data to indicate that macrophage progenitors infiltrate the embryo proper predominantly by intravascular trafficking. However, the microscopy tools applied in this paper are suited for imaging fast intravital processes and, therefore, we cannot quantify nor exclude trans-tissue migration”

We thank the reviewer for his comment. We agree that trans-tissue migration cannot be quantified or excluded by our assays. We therefore rephrased the paragraph as suggested: “We interpret our data to indicate that macrophage progenitors infiltrate the embryo predominantly by intravascular trafficking. However, the microscopy tools applied in this paper are suited for imaging fast intravital processes and, therefore, we cannot quantify nor exclude trans-tissue migration.”

Line 414: insert with between line and the. Line 415: Insert h to change were to where

We changed the sentence as suggested by the reviewer. It now reads as follows: “This is in line with the concept that EMPs travel to the embryo proper to colonize the nascent fetal liver before E10.5, where they give rise to tissue macrophages and other myeloid cells.”

To return to an issue the authors addressed in the introduction and discussion, the authors state that HSC first arise in the AGM on E10.5, and that the earlier yolk sac hematopoiesis is erythromyeloid [EMP]. That is just one side of the story. Moore and Owen first postulated yolk sac origin of full hematopoiesis and did some cell transfers to provide evidence. Weissman, Papiouannou and Gardner published in the 1970s that E8-9 yolk sac blood island cell suspensions transplanted into the yolk sacs of same age hosts distinguishable by H2 alleles and Thy 1 alleles, gave rise to both cfu-s and T cells in the adult hosts. Ueno et al demonstrated by several lineage tracing techniques that individual yolk sac blood islands do not have consistent lineage-traced endothelial cells and blood cells from the same precursor, bringing into question the identity of the hemangioblast or hemogenic endothelium as an exclusive source of hematopoietic stem/progenitor cells, or that the blood and vascular common precursor exists before the formation of blood islands. Samalkhov and Nishikawa showed by tamoxifen pulsed E7-8.5 runx 1Cre-Er mice that the resulting hematopoiesis found in adult marrow and fetal liver derived from these cells includes all blood cell lineages. Inlay et al and Mikkola et al recently provided evidence that before E10.5 yolk sac blood islands at all time points have many more phenotypically defined HSC that transplant full multilineage hematopoiesis in newborn hosts, and that these early HSC are Lyve 1+, whereas the AGM never is Lyve1+. All in all, it is obvious that the complexity of yolk sac and post yolk sac hematopoiesis from non-hematopoietic precursors during embryogenesis and fetal life is not yet clear, and that single cell tracking methods will be required to understand just how diverse the early hematopoietic cell populations are, and how many of them participate just during the embryo fetus interval vs throughout life. The finding by these authors and another group that the resident macrophages derived from yolk sac origins obviously proliferate *pari passu* with the tissues they inhabit as the tiny fetus grows to a ~25 gm adult confound the idea that all hematogenous cells are recently derived from HSC. This paper shows conclusively that *csf1R* and *cxcr3R* + precursors give rise to yolk sac origin macrophage lineage cells that, for about 4 days, emigrate massively out of the yolk sac, are found in the yolk sac associated vessels, and are found in tissues throughout the body soon after, and for as long as the mouse lives, or at least was assayed. Therefore it would seem the authors wouldn't want to bias this excellent paper with citing only one side of the HSC story, which is not really relevant to this paper.

We agree with the reviewer that yolk sac and fetal hematopoiesis is a complex subject and that further research is needed to define the diversity of early hematopoiesis. To take the above aspects into account, we added a sentence on the full hematopoietic potential of YS progenitors, and cited the respective literature as suggested (thereby reaching the limit of 70 references). The discussion section was amended as follows: "*YS hematopoiesis gives rise to multipotent cells that can develop into fetal lymphoid progenitors*¹⁻⁴."

As acknowledged by the reviewer, we already emphasized the complexity of early hematopoiesis and stressed the need for further analyses. In the discussion section we noted: "*It is possible that these cells arise from the first wave of KIT- CD16/32+ primitive progenitors*⁵. However, this population might be heterogeneous and additional markers to unequivocally identify primitive macrophages remain to be determined."

Reviewer #2:

It is now will recognized that tissue-resident macrophage populations in many adult organs are ultimately derived from hematopoiesis in the yolk sac prior to the emergence of hematopoietic stem cells (in no small part by the pioneering studies of the authors). In this revised paper, the authors have responded systematically to all of the reviewers' comments/critique with new experimental data and appropriate modifications to the text. Most importantly, they have added tracking data using Kit expression to more specifically track EMP and to distinguish them from macrophage precursors that initially arise prior to EMP in the yolk sac and go on to seed the developing brain.

The experiments are well described and the data are logically laid out. The authors provide convincing data, through images and multiple movies, that macrophage cells that arise in the yolk sac enter the bloodstream and move into the embryo to seed various organs, starting with the head region. Use of genetically-labeled cells make these elegant tracking studies possible. Evidence that macrophage precursors and EMP both use the vasculature to migrate into the embryo is the main finding of this paper.

We thank the reviewer for these positive comments.

References:

1. Ueno H and Weissman IL. Clonal analysis of mouse development reveals a polyclonal origin for yolk sac blood islands. *Dev Cell*. 2006;11:519-33.
2. Moore MA and Metcalf D. Ontogeny of the haemopoietic system: yolk sac origin of in vivo and in vitro colony forming cells in the developing mouse embryo. *British journal of haematology*. 1970;18:279-96.
3. Samokhvalov IM, Samokhvalova NI and Nishikawa S. Cell tracing shows the contribution of the yolk sac to adult haematopoiesis. *Nature*. 2007;446:1056-61.
4. Inlay MA, Serwold T, Mosley A, Fathman JW, Dimov IK, Seita J and Weissman IL. Identification of multipotent progenitors that emerge prior to hematopoietic stem cells in embryonic development. *Stem Cell Reports*. 2014;2:457-72.
5. McGrath KE, Frame JM and Palis J. Early hematopoiesis and macrophage development. *Seminars in immunology*. 2016.
6. Iqbal AJ, McNeill E, Kapellos TS, Regan-Komito D, Norman S, Burd S, Smart N, Machemer DE, Stylianou E, McShane H, Channon KM, Chawla A and Greaves DR. Human CD68 promoter GFP transgenic mice allow analysis of monocyte to macrophage differentiation in vivo. *Blood*. 2014;124:e33-44.
7. Khazen W, M'Bika J P, Tomkiewicz C, Benelli C, Chany C, Achour A and Forest C. Expression of macrophage-selective markers in human and rodent adipocytes. *FEBS Lett*. 2005;579:5631-4.
8. Panek CA, Ramos MV, Mejias MP, Abrey-Recalde MJ, Fernandez-Brando RJ, Gori MS, Salamone GV and Palermo MS. Differential expression of the fractalkine chemokine receptor (CX3CR1) in human monocytes during differentiation. *Cell Mol Immunol*. 2015;12:669-80.
9. Karlsson KR, Cowley S, Martinez FO, Shaw M, Minger SL and James W. Homogeneous monocytes and macrophages from human embryonic stem cells following coculture-free differentiation in M-CSF and IL-3. *Experimental hematology*. 2008;36:1167-75.
10. Chang GW, Davies JQ, Stacey M, Yona S, Bowdish DM, Hamann J, Chen TC, Lin CY, Gordon S and Lin HH. CD312, the human adhesion-GPCR EMR2, is differentially expressed during differentiation, maturation, and activation of myeloid cells. *Biochemical and biophysical research communications*. 2007;353:133-8.
11. van Eijk M, Aust G, Brouwer MS, van Meurs M, Voerman JS, Dijke IE, Pouwels W, Sandig I, Wandel E, Aerts JM, Boot RG, Laman JD and Hamann J. Differential expression of the EGF-TM7 family members CD97 and EMR2 in lipid-laden macrophages in atherosclerosis, multiple sclerosis and Gaucher disease. *Immunol Lett*. 2010;129:64-71.
12. Kwakkenbos MJ, Chang GW, Lin HH, Pouwels W, de Jong EC, van Lier RA, Gordon S and Hamann J. The human EGF-TM7 family member EMR2 is a heterodimeric receptor expressed on myeloid cells. *Journal of leukocyte biology*. 2002;71:854-62.